# The Impact of Women's Empowerment on the Corporate Environmental, Social, and Governance (ESG) Disclosure

**Juan Dempere ***[ID] **and Shahira Abdalla**





Business Division, Higher Colleges of Technology, Ras Al Khaimah P.O. Box 4792, United Arab Emirates
* Correspondence: jdempere@hct.ac.ae

**Abstract:** This research article examines the relationship between women's empowerment and corporate ESG disclosure variables by analyzing 10,121 publicly traded companies listed worldwide with historical ESG data available in Bloomberg from 2016 to 2020. The paper seeks to answer whether corporate gender diversity directly affects companies' ESG disclosure by using proprietary Bloomberg ESG disclosure scores and independent variables such as the female board and executive representation. Control variables, like the company's return on equity, total debt ratio, and the natural logarithm of total assets as a proxy measurement of the firm's size, are also included. Results provide evidence that policies that foster corporate gender diversity directly benefit from enhanced ESG-related disclosure, thus helping to trigger national dialogues about suitable corporate gender diversity strategies influencing firms' ESG disclosure. This paper makes a unique contribution to the literature by being the first to analyze the effects of women's empowerment on ESG disclosure using a globally representative sample. The evidence of the benefits of women's empowerment associated with corporate ESG disclosure suggests that organizations with a more gender-diverse corporate board and executive team are more likely to have higher levels of ESG disclosure, as gender diversity increases the likelihood of organizational transparency and accountability, and can lead to improved corporate value. Governments should use this evidence to implement policies promoting women's empowerment in the corporate world, ultimately leading to improved corporate ESG disclosure.

**Keywords:** ESG; environmental; social responsibility; corporate governance; gender diversity; women's empowerment; board diversity; executive diversity; ESG disclosure; sustainability

## 1. Introduction

Globally, 2021 became the year of environmental, social, and governance (ESG) investing. Kerber and Jessop [1] inform that investments in ESG-focused funds worldwide reached a record of USD 649 billion on 30 November 2021, representing a significant increase compared to the USD 542 billion and USD 285 billion invested in these funds during 2020 and 2019, respectively. They also report that ESG funds account for 10% of worldwide fund assets. According to the Sustainable Investments Institute, US companies' shareholder support for ESG initiatives increased to 32% in 2021 from 27% in 2020 and 21% in 2017 [1]. The compounding reasons include the relevance of climate change as a crucial global problem facing governments worldwide.

Simultaneously, women's empowerment has become a globally critical issue. Since the beginning of the 21st century, several countries have imposed gender quotas for corporate boards. Indeed, Kuzmina and Melentyeva [2] report national gender quotas for corporate boards in the UK (25% since 2015), France (40% since 2017), Italy (33% since 2015), Belgium (33% since 2017), Netherlands (30% since 2016), Spain (40% since 2015), and Norway (40% since 2006). Likewise, Terjesen, Lorenz, and Aquilera [3] also inform on national gender quotas for corporate boards in Finland (40% since 2005), Quebec-Canada (50% since 2011), Israel (50% since 2010), Iceland (40% since 2013), and Kenya (33% since 2010). As recently as June 2022, the European Union has decided to request publicly-traded companies with

more than 250 employees to fill 40% of their directorship positions and 33% of their senior executive positions with women starting in 2026 [4]. Equally, in 2021 the US Securities and Exchange Commission authorized the National Association of Securities Automated Quotation System (NASDAQ) to ask firms listed in NASDAQ to include at least one woman on their board of directors [4].

Similarly, in the Gulf Cooperation Council's (GCC) countries, policies have been implemented to promote gender diversity at the executive and corporate board levels. These policies include the Securities and Commodities Authority recently announcing in March 2021 that publicly traded companies in the UAE must have at least one female board director [5]. Similarly, The Ministry of Human Resources and Social Development, jointly with the Capital Market Authority, signed a memorandum of understanding to foster women's participation on Saudi publicly traded companies' boards [6]. Likewise, the Oman bourse recently announced the incorporation of two women to its seven-member board to foster board gender diversity and promote similar actions among the Omani business community [7]. According to the Association of Chartered Certified Accountants (ACCA), women represented about two percent of all board positions in the GCC countries in 2017 [8]. ACCA also informs that seventeen percent of all executive roles in the UAE are women and just seven percent in Qatar. They also allege that only thirteen percent of all Chief Executive Officers (CEOs) in the GCC region are women, while only seven percent of board chairs are women.

This paper aims to examine the relationship between women's empowerment and corporate ESG disclosure variables by analyzing 10,121 publicly traded companies listed worldwide with historical ESG data available in Bloomberg from 2016 to 2020. The study makes a unique contribution to the literature by being the first to analyze the effects of women's empowerment on ESG disclosure using a globally representative sample. The paper thus seeks to answer the question of whether corporate gender diversity has a direct impact on companies' ESG disclosure. The article tries to answer such a question by using a proprietary Bloomberg ESG disclosure score based on the extent of a company's ESG disclosure efforts, as well as the proprietary Bloomberg ESG pillars: Environmental, Social, and Governance disclosure pillars, to measure the amount of ESG-related information a company discloses publicly. Additionally, the paper examines the percentage of women on a company's board of directors and the number of female executives as a percentage of the total executives of the company as independent variables. The paper also includes control variables such as the company's return on equity, total debt ratio, and the natural logarithm of total assets as a proxy measurement of the firm's size. The paper results provide evidence that policies that foster corporate gender diversity have a direct benefit of enhanced ESG-related disclosure.

This paper employs the resource-based view (RBV) of the firm as its primary conceptual and analytical framework to explore the connection between women's empowerment and corporate ESG disclosure variables. The RBV framework posits that a firm's competitive advantage and performance are derived from its unique resources and capabilities, including tangible and intangible assets. In this context, gender diversity can be seen as a valuable resource that potentially influences corporate ESG disclosure. The RBV framework emphasizes the importance of understanding a firm's resources regarding their rarity, value, inimitability, and non-substitutability. In the case of gender diversity, the valuable insights and perspectives that women bring to decision-making processes can be considered rare and difficult to replicate. As a result, organizations with gender-diverse teams may have access to a unique pool of intangible resources that can enhance their ESG disclosure efforts.

We use independent sample *t*-tests of the top and bottom quartiles resulting from organizing our sample using our dependent variable and applying generalized linear models to examine the cross-sectional variation of our dependent variables from 2016 to 2020. Our results provide evidence that policies that foster corporate gender diversity have a direct benefit of enhanced ESG-related disclosure performance. Our results can be priceless for policymakers implementing national gender diversity policies and strategies

to optimize corporate ESG disclosure. They can also help trigger national dialogues about suitable corporate gender diversity strategies influencing firms' ESG disclosure.

There is a crucial need to explore the aspect of women's empowerment in ESG as it is a critical factor in achieving corporate transparency and accountability. Women's empowerment, defined here as giving women the power and resources for decision-making, is essential for attaining corporate ESG disclosure. By examining corporate ESG disclosure variables, the percentage of women on a company's board of directors, and the number of female executives, this research article seeks to answer whether women's empowerment proxied by corporate gender diversity directly impacts companies' ESG disclosure. The evidence of the benefits of women's empowerment associated with corporate ESG disclosure suggests that organizations with a more gender-diverse corporate board and executive team are more likely to have higher levels of ESG disclosure. Governments should use this evidence to implement policies promoting women's empowerment in the corporate world, ultimately leading to improved corporate ESG disclosure.

*Literature Review*

ESG performance and disclosure are both essential concepts in the world of corporate responsibility and sustainability. Both terms refer to how companies manage their environmental, social, and governance obligations to their stakeholders, but there are significant differences between them.

First, ESG performance measures how well a company meets its ESG obligations. It focuses on outcomes and results, such as the company's carbon footprint or workforce diversity. It is assessed through external ratings, such as those provided by Standard & Poor's, and internal measurements and metrics, such as the company's sustainability reporting. Government interest in ESG performance is generally limited, as governments need more resources or capability to measure or enforce ESG performance. Managerial interest, however, is high since ESG performance is often used as a metric for executive compensation and other performance-based rewards. The impact of ESG performance is mainly positive, as companies that perform well on their ESG obligations are likely to be more responsible and sustainable.

ESG disclosure, on the other hand, is a measure of how well a company is communicating its ESG obligations to its stakeholders. It focuses on the company's transparency and communication, such as how much information it discloses in its sustainability reports and other documents. It is assessed through external ratings, such as those provided by Bloomberg, and internal measurements and metrics, such as the company's sustainability reporting. Government interest in ESG disclosure is generally high, as governments often require companies to disclose certain information and are increasingly creating regulations and standards around ESG disclosure. Managerial interest is also high since ESG disclosure can increase public awareness and trust in the company, positively impacting its reputation and financial performance. The impact of ESG disclosure is mainly positive, as it can lead to increased public confidence in the company and greater accountability and transparency.

Therefore, ESG performance and ESG disclosure are critical concepts in corporate responsibility and sustainability. While they are related, they are not the same. ESG performance focuses on actual outcomes and results, while ESG disclosure focuses on the company's transparency and communication. Government interest and managerial interest in each are different, and the impacts of each are distinct. Companies should strive to excel in ESG performance and ESG disclosure to be considered responsible and sustainable.

The elucidation above is essential to grasp the distinctive contribution of the current article in comparison to prior studies. The previous articles outlined below analyzing the relationship between ESG performance and corporate gender diversity supply information that is advantageous yet conceptually dissimilar to our examination. We investigate the relationship between ESG performance and corporate gender diversity, and this aspect of investigation has its unique characteristics, as the theoretical exposition above supports.

A related theme in the academic ESG-related literature includes studies about the relationship between corporate gender diversity and ESG performance with mixed results. Some articles find a positive relationship between board gender diversity and ESG performance [9–11], while others find a negative relationship between these variables [12]. Other studies find a significant positive relationship between board gender diversity and ESG performance. Still, they are limited to individual countries like the United States [13], Italy [14], Germany & Austria [15], China [16], Canada [17], India [18], etcetera. Alternatively, some studies have focused on specific regions [19,20] or a particular ESG-related disclosure metric (e.g., voluntary carbon emissions [21]). The limited analysis of previous articles also includes emphasis on some specific industry sectors: banks [22,23], oil and gas [13], transport and logistics [24], etcetera.

There are also some previous articles about corporate gender diversity and ESG disclosure. Indeed, Gurol and Lagasio [25] investigate the relationship between banks' board structure and sustainability performance by analyzing a sample of 35 European banks listed at the EURO STOXX 600. Results show that having a larger board with a high proportion of women and independent members has a positive effect on ESG disclosure scores. Likewise, Wan Mohammad, Zaini, and Md Kassim [26] study the impact of women on board and firms' competitive advantage on firms' ESG disclosure. Using 332 firm-year observations of 65 firms in Bursa Malaysia, the authors find that women on board encourage ESG and environmental disclosures. Disli, Yilmaz, and Mohamed [27] examine the impact of board attributes on the sustainability performance of 439 publicly-listed non-financial companies in 20 emerging countries from 2010 to 2019. They finding a positive relationship between board gender diversity and sustainability performance across various sustainability indicators.

In the same way, Khemakhem, Arroyo, and Montecinos [28] analyze the relationship between gender diversity on the board of directors and its committees and the environmental, social, and governance (ESG) disclosure of Canadian-listed companies. They find that board gender diversity can positively and significantly affect companies' ESG disclosure, with committees' gender diversity having a more substantial influence than the board itself. Equally, Manita et al. [29] examine the relationship between corporate debt-like compensation and excess cash holdings, using the ESG disclosure score as a proxy for corporate social responsibility (CSR). Through a sample of 379 firms, the authors find no significant relationship between board gender diversity and ESG disclosure; however, they suggest that further research should extend the time and space parameters. However, they argue for the feminization of corporate boards, as transparency correlates positively with this. Correspondingly, Bravo and Reguera-Alvarado [30] investigate the effect of female representation on audit committees (ACs) on the quality of ESG disclosure. They find that more female AC members lead to more comprehensive and relevant ESG reporting.

Similarly, Jizi, Nehme, and Melhem [31] examine the relationship between board gender diversity and firm social engagement in GCC countries. They find that the role of women on boards in prompting firms' social agenda and enhancing the level of sustainability reporting. Likewise, Cucari et al. [32] investigate the association between ESG disclosure and board diversity in Italian-listed companies and find that a firm's CSR disclosure is associated with independent director and committee CSR. In contrast, the proportion of women directors is negatively correlated. Correspondingly, Wasiuzzaman and Subramaniam [33] investigate the role board gender diversity plays in the quality of ESG disclosures among energy firms across 48 developed and developing countries from 2004 to 2016. Their findings indicate that female directors generally enhance ESG disclosure quality, however, this trend is particularly pronounced within developed nations. Equally, Nicolò et al. [34] study the impact of boardroom gender diversity on ESG disclosure practices in European listed firms and find that the presence of women directors on boards enhances ESG disclosure, both at the overall and specific ESG score level.

Correspondingly, Qureshi et al. [35] examine an extensive panel data set of 812 European firms to investigate the impact of sustainability disclosure and female representation

on boards on firm value and find that sustainability disclosure, best management practices, stakeholder trust, and board gender diversity all have a positive effect on firm value. Furthermore, they find that firms in sensitive industries have superior social and governance performance, and firms with higher female representation on their boards present better ESG disclosure. Similarly, Wasiuzzaman et al. [36] investigate the impact of board gender diversity on the transparency of ESG disclosure in Malaysia and find that ESG disclosure scores are higher with an increased presence of female directors. Additionally, Zumente and Lāce [37] analyze the association between board diversity and ESG disclosure for companies listed on the NASDAQ OMX Baltic Stock exchange and find that companies with larger boards and female representation on supervisory boards have higher non-financial disclosure scores. Lastly, Wan Ismail et al. [38] investigate the relationship between gender diversity in the boardroom and corporate cash holdings, as well as the moderating effects of investor protection, and find that board gender diversity has a negative association with corporate cash holdings. This effect is weaker in countries with higher levels of investor protection.

The use of financial-related control variables in our study is justified by the relevance of the relationship between financial performance and ESG scores in the ESG-related academic literature with mixed results [39,40]. Some articles find a non-significant relationship between these two factors [41–43], while others find a negative correlation [44]. Some manuscripts also report a positive impact of ESG activities on firm market value, but they lack consensus about which ESG factors have the most significant influence [45,46].

The current article builds upon the literature review of previous academic papers by providing evidence of the benefits of women's empowerment associated with corporate ESG disclosure. This building is done distinctively by being the first to analyze the effects of gender diversity on ESG disclosure using a globally representative sample of 10,121 companies worldwide with historical ESG data available in Bloomberg from 2016 to 2020. This sample is highly descriptive of the global economy as it contains companies from 92 countries, with the largest number of firms from the world's three most prominent and influential economies: the United States, Japan, and China. Considering a global sample, the current article provides evidence of a universal association between gender diversity and ESG disclosure, regardless of any cultural-related factors that might influence previous research. This fact is an essential distinction of earlier articles that have focused on the effects of gender diversity and ESG disclosure in individual countries or regions, which could be affected by cultural-related factors.

The variables analyzed include the percentage of women on the board of directors, the number of female executives as a percentage of the total executives of the company, a company's annual return on equity, the total debt ratio, and the natural logarithm of total assets. These variables are analyzed in the context of corporate ESG disclosure using a novelty theoretical approach based on the firm's RBV. This analysis suggests that organizations have access to both tangible and intangible resources that can be leveraged to create value.

The current article also attempts to fill the gap in the literature by providing evidence of the positive impacts of policies that foster corporate gender diversity on ESG disclosure. This attempt offers theoretical and practical insights to managers, investors, government officers, professionals, etcetera, on the importance of gender diversity for corporate ESG disclosure. This article's findings suggest that policies promoting gender diversity at the executive and board levels should be implemented to enhance corporate ESG disclosure. The results also indicate that gender-diverse teams may be more likely to create a culture of ethical conduct, which is essential for adequate ESG disclosure.

## 2. Materials and Methods

We retrieved ESG data from the Bloomberg database. Our dependent variables include a proprietary Bloomberg ESG disclosure score (Y1) based on the extent of a company's ESG disclosure efforts. The metric ranges from zero (0) for firms that do not disclose any of the

ESG data points comprised by this metric to one hundred (100) for those firms that make known every data point. Our dependent variables include the proprietary Bloomberg ESG disclosure pillars: Environmental, Social, and Governance. These disclosure pillars' scores also range from zero (0) for businesses that do not release any of the Environmental, Social, or Governance data points included in the corresponding pillars to one hundred (100) for those companies that release every single data point included in the individual pillar. These dependent variables measure the amount of publicly disclosed ESG-related information a company discloses but does not measure a firm's ESG-related performance on any data point.

Our sample of 10,121 publicly traded companies listed worldwide with historical ESG data available in Bloomberg from 2016 to 2020 highly represents the global economy. The sample contains companies from 92 countries, with the largest number of firms from the US, with 2286 companies representing 23.41%, followed by Japan (1820 firms or 18.64%) and China (1259 firms or 12.89%). These constituents indicate the current global economic landscape, with these three countries being the world's largest and most influential economies. The sample also includes Taiwan (354 firms or 3.63%), India (325 firms or 3.33%), Germany (274 firms or 2.81%), South Korea (222 firms or 2.27%), Canada (221 or 2.26%), Australia (193 firms or 1.98%), and the rest of the world with 2301 firms or 23.57%. These components reflect each country's global capital market share, with the sample proportionally representing the global economic landscape. Therefore, this sample is highly representative of publicly traded companies worldwide and provides an accurate representation of the global economy. Figure 1 below provides a graphical representation of the countries with the most significant numbers of firms in our sample.

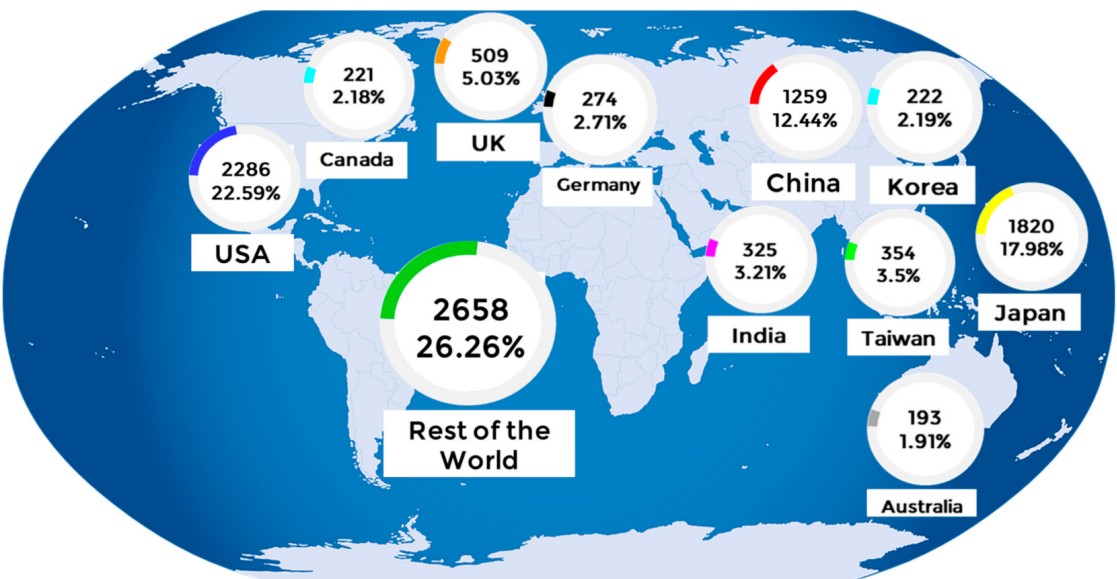

**Figure 1.** Sample's World Distribution.

The Bloomberg database calculates the general ESG-related disclosure scores and constituent pillars using hundreds of data points from firm public disclosures. The ESG disclosure score is based on the corresponding environmental, social, and corporate governance disclosure pillars' values. The ESG disclosure score is a relative sum of the pillars' weights, which vary according to the company's industry sector. The environmental disclosure pillar (Y2) includes many data points like ISO 14,000 certifications, the number of spills, the number & amounts of ecological fines, emissions ($CO_2$, CO, Methane, particles, ODS, etc.), renewable energy use, energy consumption, water use, recycled materials, waste materials, waste management and recycling, environmental supply chain management, biodiversity impact reduction, environmental & eco-designed products, environmental R&R expenditures, animal testing, etcetera. The social disclosure pillar (Y3) includes data

points associated with employee turnover, strikes, women managers, number of accidents, occupational diseases, employee fatalities, policies on child labor, human rights, freedom of association, forced labor, bribery & corruption policy, business ethics policy, total donations, political contribution, corporate health & safety policy, revenues from alcohol, gambling, tobacco, weapons, weapons, pornography, etcetera. The governance disclosure pillar (Y4) includes data points related to board cultural and gender diversity, organization and independence of board committees, corporate policies on the board's functions, size, independence, diversity, experience, etc., board member affiliation, re-election, term duration, compensation, shareholders rights policy, voting cap, minimum shares voting requirements, poison pills, severance agreements, auditor tenure, supermajority vote requirements, ESG reporting scope, etcetera.

Our independent variables include the percentage of women on the board of directors (X1) as informed by each company from the Bloomberg database. Similarly, we obtained the number of female executives as a percentage of the total executives of the company (X4). Equally, we include in our analysis some control variables comprising the company's annual return on equity (X2), total debt ratio (X3), and the natural logarithm of total assets (X5). These control variables have been utilized in previous research works. Indeed, Wahyuningrum, Oktavilia, and Utami [47] study the effect of profitability, leverage, firm size, industry type, and gender diversity on sustainability reports by the Sustainable Development Goals. Using a sample of 112 companies listed on the Indonesia Stock Exchange, they find that leverage, industry type, and gender diversity significantly affect sustainability reports.

A company's annual return on equity (X2), total debt ratio (X3), and the natural logarithm of total assets (X5) are essential control variables for this study because they can provide a clear indicator of the financial condition of the company and its ability to support ESG initiatives. The return on equity and total debt ratio offers insight into a company's profitability and financial stability. In contrast, the natural logarithm of total assets can be used to measure the size of the company and its available resources. Including these control variables allows us to isolate the effects of X1 and X4 while still considering the potential impacts of other factors on the dependent variables. Considering these financial indicators, the study's results will be more meaningful and provide a more accurate assessment of any relationship between ESG disclosure scores and gender diversity at the board and executive levels.

We employed generalized linear models to assess the cross-sectional variation of our dependent variables from 2016 to 2020. Our models were determined by a linear predictor $\eta i = \beta 0 + \beta 1 IVAR1i + \cdots + \beta p IVARpi$; as well as two identities: a link equation describing the mean $(DVi) = \mu i$, which is a function of the linear predictor $(\mu i) = \eta i$; and a variance function defining how the variance, var(Yi) relies on the mean var$(DVi) = \phi V(\mu)$, where the distribution factor $\phi$ is a constant. For our general linear models, we have $\varepsilon = (0, \sigma 2)$, where the linear predictor $\eta i$ has previously been defined, the link equation $g(\mu i) = \mu i$, and the variance equation $V(\mu i) = 1$. Additionally, we implemented logarithmic transformations to some of our control variables when utilizing the regression models described above.

Using the methodology described above, we tested the following hypotheses:

**H1:** *There is a positive relationship between ESG disclosure scores and the number of female directors on a firm's board.*

**H2:** *There is a positive relationship between ESG disclosure scores and the percentage of female executives in a firm.*

Alternatively, in this study, we addressed the following research questions:

*RQ1: What is the relationship between ESG disclosure scores and the number of female directors on a firm's board?*

*RQ2: What is the relationship between ESG disclosure scores and the percentage of female executives in a firm?*

These hypotheses and research questions guided our data analysis and the interpretations of the results. The following generalized linear model was used to test the cross-sectional variation of our dependent variables facilitating the replication of our study: $Yi = \beta0 + \beta1X1 + \beta2X2 + \beta3X3 + \beta4X4 + \beta5X5$, where *i* identifies our four dependent variables Bloomberg' ESG disclosure score (Y1), Environmental disclosure pillar score (Y2), Social disclosure pillar score (Y3), and Governance disclosure pillar score (Y4). The model predicts the proprietary Bloomberg dependent variables Y1–Y4 based on the percentage of women on the board of directors (X1), the company's annual return on equity (X2), total debt ratio (X3), the percentage of female executives (X4) and the natural logarithm of total assets (X5) as a proxy measurement of the firm's size. According to hypotheses H1 & H2, we expect the sign of the beta coefficients of X1 and X4 to be positive.

### 3. Results

Chart 1 shows the annual averages of our dependent variables for our sample of 10,121 companies from 2016 to 2020. We can verify in the chart that the lowest average disclosure scores are those of the environmental disclosure pillar. This result may suggest that the ecological dimension is highly sensitive from a legal and corporate image viewpoint. Additionally, measuring the environmental disclosure pillar's data points requires technical capabilities more sophisticated than the other two dimensions (e.g., emissions measurement, energy efficiency, etc.). Therefore, it seems reasonable to consider this dimension the least transparent from a corporate disclosure perspective.

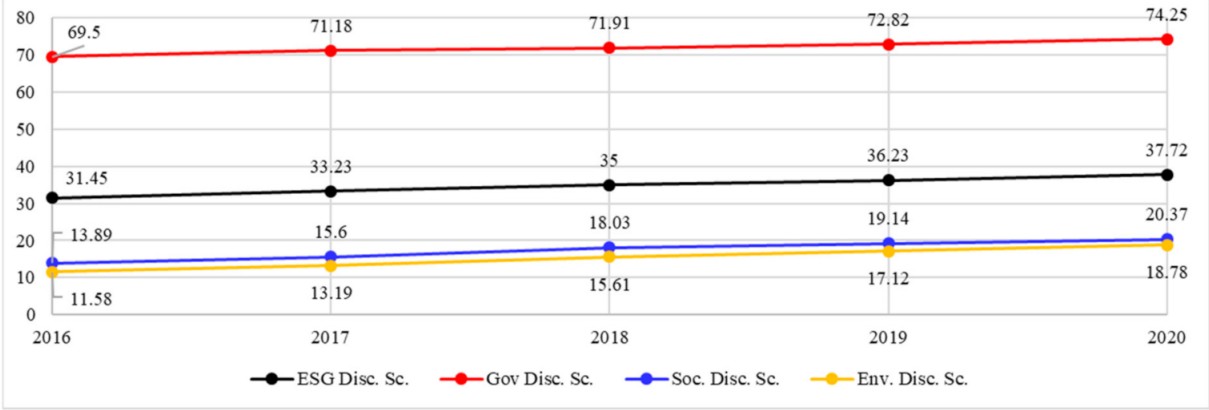

**Chart 1.** Scores of company averages of ESG disclosure scope.

Similarly, Chart 1 shows that the highest average disclosure scores are those of the governance disclosure pillar. This result is consistent with the nature of the data points associated with this pillar, in the sense that all variables in this pillar are easy to measure (e.g., board diversity, employee turnover, etc.). The companies' periodic financial statements must also disclose many of these metrics. However, Chart 1 shows that the annual averages of all ESG disclosure scores and their constituent pillars have a rising yearly trend. This result suggests the growing significance of ESG-related corporate disclosure requirements.

Chart 2 shows the annual values of our independent variables for gender diversity at the board and executive levels. The yellow bars indicate the yearly average percentage of women on the board of directors as informed by each company. Similarly, the red bars show each company's annual average rate of female executives. We can verify that women's empowerment at the board level is inferior to that at the executive level. However, the chart shows a growing trend for both dimensions of women's empowerment analyzed in our study. This result reflects the relevance of this trend worldwide.

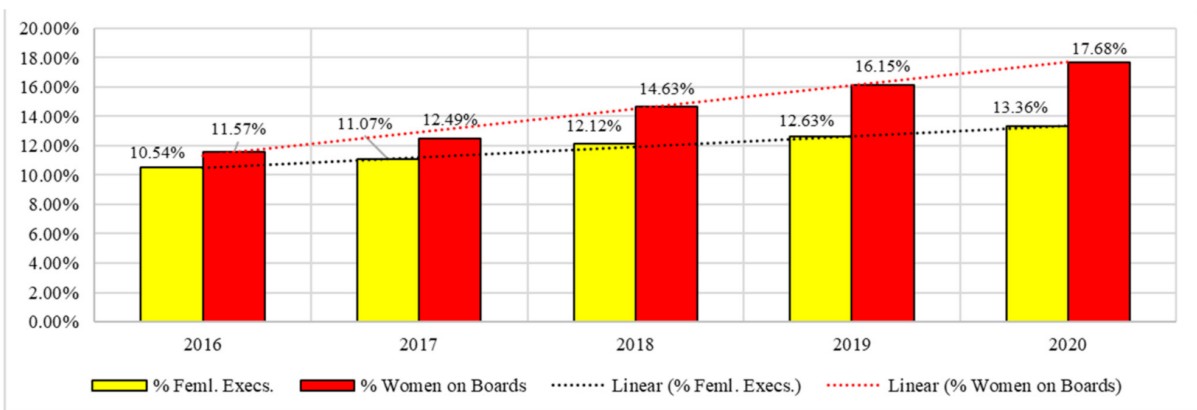

**Chart 2.** Gender diversity at the board and executive levels.

For analytical purposes, we organized our sample by the gender diversity at the board and executive level, from the lowest (zero women's representation) to the highest values, and identified the first and fourth quartiles. We then compared the average figures of our dependent variables for each quartile. Tables 1 and 2 show the independent samples test of our dependent variables for our sample of 10,121 companies organized by gender diversity at the board and executive levels. The statistically significant results show that companies with no female representation at the board level (0%) have consistently superior scores for the general ESG disclosure and all its constituent pillars. This result is the same for every analyzed year from 2016 to 2020.

**Table 1.** Independent Samples Test—Percentage of Women on the Board of Directors (X1).

| Year | $Q_{1/4}/p/t$ | X1 | Y1 | Y2 | Y3 | Y4 |
|---|---|---|---|---|---|---|
| 2020 | Q1 | 0.00% | 31.2383 | 66.1244 | 15.6658 | 12.6061 |
| | Q4 | 37.34% | 44.3854 | 78.9761 | 26.9528 | 27.3644 |
| | $t$-sta. | −223.5 | −36.1 | −31.673 | −31.761 | −26.595 |
| | $p$-val. | [0.00] **** | [0.00] **** | [0.00] **** | [0.00] **** | [0.00] **** |
| 2019 | Q1 | 0.00% | 31.9264 | 67.042 | 15.93 | 13.4534 |
| | Q4 | 35.70% | 42.8805 | 77.736 | 25.66 | 25.608 |
| | $t$-sta. | −203.7 | −28.6 | −25.017 | −26.29 | −20.604 |
| | $p$-val. | [0.00] **** | [0.00] **** | [0.00] **** | [0.00] **** | [0.00] **** |
| 2018 | Q1 | 0.00% | 31.3018 | 66.8767 | 14.8833 | 12.7548 |
| | Q4 | 33.94% | 41.9479 | 77.0806 | 24.7993 | 24.2751 |
| | $t$-sta. | −189.411 | −27.716 | −23.077 | −27.124 | −19.417 |
| | $p$-val. | [0.00] **** | [0.00] **** | [0.00] **** | [0.00] **** | [0.00] **** |
| 2017 | Q1 | 0.00% | 30.7827 | 67.3086 | 13.4643 | 11.8716 |
| | Q4 | 30.64% | 39.6389 | 76.3586 | 21.7139 | 21.0297 |
| | $t$-sta. | −161.99 | −22.686 | −21.025 | −20.977 | −14.725 |
| | $p$-val. | [0.00] **** | [0.00] **** | [0.00] **** | [0.00] **** | [0.00] **** |
| 2016 | Q1 | 0.00% | 29.148 | 66.2855 | 11.4017 | 10.0304 |
| | Q4 | 28.82% | 37.5495 | 75.1225 | 19.5488 | 18.4381 |
| | $t$-sta. | −152.99 | −22.154 | −19.254 | −21.006 | −14.097 |
| | $p$-val. | [0.00] **** | [0.00] **** | [0.00] **** | [0.00] **** | [0.00] **** |

Notes: **** denote statistical significance at the 0.1% significance level. The table contains z-statistic and their corresponding *p*-values below in parentheses. The variables on the table are the percentage of women on the board of directors (X1), Bloomberg's ESG disclosure score (Y1), Environmental disclosure pillar score (Y2), Social disclosure pillar score (Y3), and Governance disclosure pillar score (Y4).

**Table 2.** Independent Samples Test—Female Executives, as a % of Total Executives (X4).

| Year | $Q_{1/4}$/p/t | X4 | Y1 | Y2 | Y3 | Y4 |
|------|-----------|-----|-----|-----|-----|-----|
| | Q1 | 0.00% | 25.9226 | 64.2813 | 10.6791 | 3.2219 |
| 2020 | Q4 | 36.55% | 41.2673 | 77.2469 | 23.9604 | 23.0888 |
| | t-sta. | −142.56 | −50.463 | −33.364 | −41.559 | −41.903 |
| | p-val. | [0.00] **** | [0.00] **** | [0.00] **** | [0.00] **** | [0.00] **** |
| | Q1 | 0.00% | 38.1863 | 69.8528 | 23.0092 | 22.557 |
| 2019 | Q4 | 35.39% | 40.0258 | 76.3086 | 22.8891 | 21.5961 |
| | t-sta. | −135.8 | −4.555 | −14.618 | 0.302 | 1.559 |
| | p-val. | [0.00] **** | [0.00] **** | [0.00] **** | [0.763] | [0.119] |
| | Q1 | 0.00% | 37.4472 | 69.1206 | 22.3019 | 21.8424 |
| 2018 | Q4 | 34.37% | 38.7496 | 76.0878 | 21.2573 | 19.3124 |
| | t-sta. | −123.523 | −3.227 | −15.251 | 2.686 | 4.08 |
| | p-val. | [0.00] **** | [0.001] *** | [0.00] **** | [0.007] *** | [0.00] **** |
| | Q1 | 0.00% | 36.2463 | 69.2413 | 20.2963 | 19.8071 |
| 2017 | Q4 | 32.34% | 35.9156 | 75.1637 | 17.7671 | 15.1711 |
| | t-sta. | −116.867 | 0.811 | −12.865 | 6.017 | 7.256 |
| | p-val. | [0.00] **** | [0.417] | [0.00] **** | [0.00] **** | [0.00] **** |
| | Q1 | 0.00% | 34.6424 | 68.1274 | 18.4817 | 18.4594 |
| 2016 | Q4 | 31.43% | 34.4514 | 74.3537 | 15.9419 | 13.5161 |
| | t-sta. | −114.446 | 0.475 | −12.826 | 6.012 | 7.799 |
| | p-val. | [0.00] **** | [0.417] | [0.00] **** | [0.00] **** | [0.00] **** |

Notes: **** and *** denote statistical significance at the 0.1% and 1% significance levels, respectively. The table contains z-statistic and their corresponding *p*-values below in parentheses. The variables on the table are the female executives as a percentage of total executives (X4), Bloomberg's ESG disclosure score (Y1), Environmental disclosure pillar score (Y2), Social disclosure pillar score (Y3), and Governance disclosure pillar score (Y4).

Tables 1 and 2 also show mixed results for companies with no executive gender diversity versus those with some percentage of executive women. The statistically significant results show that companies with no executive women (0%) have consistently low disclosure scores for the general ESG metric, but only for 2020–2018, since our results are insignificant for this dependent variable for 2017 and 2018. Similarly, Tables 1 and 2 show that companies with gender diversity at the executive level have consistently low scores for Governance disclosure every year between 2016 and 2020.

Our results in Tables 1 and 2 show that companies with no female representation at the executive level have low Social and Environmental disclosure scores but only in 2020. Our results were insignificant for these pillars in 2019 but significant and with inverse direction during 2016–2018. In other words, those companies with no executive women from 2016 to 2018 exhibited superior Social and Environmental disclosure scores. These mixed results suggest that the influence of gender diversity at the corporate executive level changes over time.

Tables 3 and 4 show our cross-sectional analysis for the ESG disclosure score as the dependent variable for 2026–2020. Using generalized linear models, we find the same results as those in Tables 1 and 2: a positive and significant relationship between female representation at the board level and the general ESG disclosure score, including all of its constituent pillars, and for all the analyzed years.

Our results for Tables 3 and 4 are also consistent with those of Tables 1 and 2 for those companies with no women at the executive level. The positive and significant relationship between the ESG disclosure score and the executive female representation is only significant for 2020–2018 but insignificant for 2017–2016. Similarly, the relationship between executive gender diversity and Governance disclosure scores is positive and significant every year from 2016 to 2020.

Finally, the relationship between women's empowerment at the executive level and the Social and Environmental disclosure scores are positive and significant in 2020 and 2019. Only in 2018 is the relationship between Social disclosure scores and executive gender diversity positive and significant. The remaining relationships for 2018, 2017, and 2016 are insignificant, with mixed signs.

**Table 3.** Cross-Sectional Analysis Results for ESG (Y1) & Environmental (Y2) Disclosures.

| | Years | 2020 | 2019 | 2018 | 2017 | 2016 |
|---|---|---|---|---|---|---|
| **ESG Disclosure (Y1)** | **C** | −9.15 | −6.79 | −3.87 | 3.03 | 3.90 |
| | *z-sta.* | −7.05 | −5.25 | −2.89 | 2.32 | 2.95 |
| | *p*-val. | (0.0) **** | (0.0) **** | (0.0) **** | (0.02) ** | (0.0) **** |
| | **X1** | 29.90 | 27.72 | 27.95 | 27.41 | 26.68 |
| | *z-sta.* | 20.22 | 17.32 | 17.29 | 15.57 | 13.21 |
| | *p*-val. | (0.0) **** | (0.0) **** | (0.0) **** | (0.0) **** | (0.0) **** |
| | **X2** | 0.51 | 0.53 | 0.45 | 0.40 | 0.37 |
| | *z-sta.* | 2.51 | 2.39 | 2.92 | 3.60 | 4.01 |
| | *p*-val. | (0.012) ** | (0.017) ** | (0.004) *** | (0.0) **** | (0.0) **** |
| | **X3** | 8.83 | 7.09 | 6.04 | 5.65 | 4.66 |
| | *z-sta.* | 7.48 | 6.62 | 8.04 | 7.57 | 6.53 |
| | *p*-val. | (0.0) **** | (0.0) **** | (0.0) **** | (0.0) **** | (0.0) **** |
| | **X4** | 5.06 | 4.80 | 3.67 | 1.81 | 1.29 |
| | *z-sta.* | 4.17 | 3.78 | 2.83 | 1.11 | 0.70 |
| | *p*-val. | (0.0) **** | (0.0) **** | (0.005) *** | (0.267) | (0.486) |
| | **X5** | 1.74 | 1.65 | 1.52 | 1.18 | 1.10 |
| | *z-sta.* | 30.61 | 28.76 | 25.68 | 20.21 | 18.56 |
| | *p*-val. | (0.0) **** | (0.0) **** | (0.0) **** | (0.0) **** | (0.0) **** |
| **Environmental Disclosure (Y2)** | **C** | −53.77 | −51.97 | −49.57 | −44.92 | −42.85 |
| | *z-sta.* | −29.30 | −29.94 | −27.16 | −23.44 | −22.90 |
| | *p*-val. | (0.0) **** | (0.0) **** | (0.0) **** | (0.0) **** | (0.0) **** |
| | **X1** | 35.47 | 32.87 | 33.06 | 33.06 | 31.23 |
| | *z-sta.* | 17.77 | 15.84 | 15.32 | 13.11 | 11.18 |
| | *p*-val. | (0.0) **** | (0.0) **** | (0.0) **** | (0.0) **** | (0.0) **** |
| | **X2** | 3.29 | 3.10 | 2.41 | 1.39 | 0.64 |
| | *z-sta.* | 5.37 | 6.33 | 4.31 | 2.82 | 1.99 |
| | *p*-val. | (0.0) **** | (0.0) **** | (0.0) **** | (0.005) *** | (0.047) ** |
| | **X3** | 14.06 | 10.46 | 7.80 | 5.88 | 3.69 |
| | *z-sta.* | 8.27 | 7.14 | 6.47 | 4.87 | 3.23 |
| | *p*-val. | (0.0) **** | (0.0) **** | (0.0) **** | (0.0) **** | (0.0) **** |
| | **X4** | 5.15 | 3.96 | 1.39 | −1.10 | −2.43 |
| | *z-sta.* | 2.97 | 2.21 | 0.75 | −0.47 | −0.94 |
| | *p*-val. | (0.003) *** | (0.03) ** | (0.452) | (0.638) | (0.349) |
| | **X5** | 2.78 | 2.74 | 2.65 | 2.38 | 2.27 |
| | *z-sta.* | 33.77 | 34.71 | 31.70 | 27.38 | 26.50 |
| | *p*-val. | (0.0) **** | (0.003) *** | (0.0) **** | (0.0) **** | (0.005) *** |

Notes: ****, *** and ** denote statistical significance at the 0.1%, 1% and 5% significance levels, respectively. The table contains z-statistic and their corresponding *p*-values below in parentheses. The variables on the table are the percentage of women on the board of directors (X1), the number of female executives as a percentage of the total executives of the company (X4), the company's annual return on equity (X2), total debt ratio (X3), and the natural logarithm of total assets (X5).

**Table 4.** Cross-Sectional Analysis Results for Social (Y3) & Governance (Y4) Disclosures.

| | Years: | 2020 | 2019 | 2018 | 2017 | 2016 |
|---|---|---|---|---|---|---|
| **Social Disclosure (Y3)** | **C** | −19.97 | −18.07 | −16.90 | −18.23 | −19.46 |
| | *z-sta.* | −16.58 | −15.59 | −14.51 | −15.08 | −15.76 |
| | *p*-val. | (0.0) **** | (0.0) **** | (0.0) **** | (0.0) **** | (0.0) **** |
| | **X1** | 26.82 | 25.71 | 26.59 | 26.22 | 26.39 |
| | *z-sta.* | 20.27 | 18.56 | 18.28 | 14.96 | 12.86 |
| | *p*-val. | (0.0) **** | (0.0) **** | (0.0) **** | (0.0) **** | (0.0) **** |
| | **X2** | 1.75 | 1.86 | 1.54 | 1.33 | 0.95 |
| | *z-sta.* | 5.79 | 7.23 | 4.64 | 4.39 | 4.40 |
| | *p*-val. | (0.0) **** | (0.0) **** | (0.0) **** | (0.0) **** | (0.0) **** |
| | **X3** | 8.39 | 6.55 | 5.02 | 4.44 | 3.76 |
| | *z-sta.* | 7.48 | 6.61 | 6.74 | 5.83 | 4.92 |
| | *p*-val. | (0.0) **** | (0.0) **** | (0.0) **** | (0.0) **** | (0.0) **** |

**Table 4.** *Cont.*

| | Years: | 2020 | 2019 | 2018 | 2017 | 2016 |
|---|---|---|---|---|---|---|
| **Social Disclosure (Y3)** | X4 | 6.06 | 4.74 | 3.16 | 1.67 | 1.14 |
| | z-sta. | 4.93 | 3.68 | 2.38 | 0.94 | 0.55 |
| | p-val. | (0.0) **** | (0.0) **** | (0.02) ** | (0.346) | (0.583) |
| | X5 | 1.47 | 1.40 | 1.35 | 1.33 | 1.33 |
| | z-sta. | 27.28 | 26.71 | 25.56 | 25.14 | 23.99 |
| | p-val. | (0.0) **** | (0.0) **** | (0.0) **** | (0.0) **** | (0.0) **** |
| | C | 51.07 | 54.83 | 59.28 | 74.86 | 76.00 |
| | z-sta. | 22.92 | 23.91 | 24.43 | 35.79 | 34.69 |
| | p-val. | (0.0) **** | (0.0) **** | (0.0) **** | (0.0) **** | (0.0) **** |
| **Governance Disclosure (Y4)** | X1 | 26.48 | 23.77 | 23.11 | 22.08 | 22.11 |
| | z-sta. | 14.81 | 11.97 | 11.42 | 11.95 | 10.68 |
| | p-val. | (0.0) **** | (0.0) **** | (0.0) **** | (0.0) **** | (0.0) **** |
| | X2 | 0.17 | 0.28 | 0.30 | 0.47 | 0.54 |
| | z-sta. | 0.58 | 1.30 | 1.37 | 3.66 | 5.90 |
| | p-val. | (0.56) | (0.19) | (0.17) | (0.0) **** | (0.0) **** |
| | X3 | 5.06 | 4.72 | 5.78 | 6.95 | 6.69 |
| | z-sta. | 4.97 | 4.45 | 6.47 | 7.67 | 7.30 |
| | p-val. | (0.0) **** | (0.0) **** | (0.0) **** | (0.0) **** | (0.0) **** |
| | X4 | 4.01 | 5.75 | 6.49 | 5.10 | 5.30 |
| | z-sta. | 3.16 | 4.20 | 4.52 | 3.20 | 2.83 |
| | p-val. | (0.002) *** | (0.0) **** | (0.0) **** | (0.0) **** | (0.005) *** |
| | X5 | 0.76 | 0.58 | 0.37 | −0.30 | −0.39 |
| | z-sta. | 8.76 | 6.37 | 3.89 | −3.50 | −4.32 |
| | p-val. | (0.0) **** | (0.0) **** | (0.0) **** | (0.0) **** | (0.0) **** |

Notes: ****, *** and ** denote statistical significance at the 0.1%, 1% and 5% significance levels, respectively. The table contains z-statistic and their corresponding *p*-values below in parentheses. The variables on the table are the percentage of women on the board of directors (X1), the number of female executives as a percentage of the total executives of the company (X4), the company's annual return on equity (X2), total debt ratio (X3), and the natural logarithm of total assets (X5).

Our results suggest that gender diversity is essential when studying corporate ESG disclosure. Companies with higher gender diversity at both board and executive levels will likely achieve higher scores in all ESG disclosure pillars. However, the influence of gender diversity at the executive level may change over time. Therefore, further research is needed to study the effect of gender diversity on ESG disclosure scores in different contexts and perspectives.

## 4. Discussion

Our findings suggest that policies that foster corporate gender diversity benefit corporate ESG disclosure. To ensure that corporate ESG disclosure is enhanced, governments should implement policies that promote gender diversity at the executive and board levels. These can include initiatives such as setting targets for the minimum representation of women in senior positions and on boards, introducing quotas for women on boards, or offering incentives to companies that meet gender diversity targets. Moreover, governments should also promote the education and training of female professionals to increase their representation in the corporate world.

The concept of corporate gender diversity has been gaining momentum in recent years, with an increasing focus on environmental, social, and governance (ESG) disclosure. Recent evidence suggests that gender diversity is an essential factor in organizational performance. Indeed, according to S&P Global [48], investors are pressing public companies to improve diversity in director ranks, reflecting a greater recognition of the importance of ESG matters. This pressure made investors consider gender diversity and equity when evaluating how firms may respond to ESG risks and opportunities. Companies are thus coming under external pressure from investors, activists, and potential customers & employees to increase

the representation of women in senior roles and equal compensation and mobility for women and people of color.

S&P Global Market Intelligence's study, When Women Lead, Firms Win [49], discovered that firms with female CFOs are more profitable and have provided superior stock performance compared to the market average; additionally, firms with greater gender diversity on their board of directors have been more lucrative and prominent than those with less. This superior performance has led to governments and regulators closely monitoring companies' female representation, with some countries introducing laws requiring a certain amount of female representation on boards.

Despite progress, women remain underrepresented in higher positions, and discrimination and misconduct remain common, particularly in the oil and gas sector. As the benefits of gender diversity become evident, public companies will continue to face investor pressure to act accordingly. The benefits of gender diversity mentioned above result from the fact that gender-diverse boards are more likely to consider a broader range of perspectives and experiences when making decisions. This diversity, in turn, can lead to more effective decision-making and better outcomes for the organization.

The importance of gender diversity at the corporate board and executive levels is also related to corporate ESG disclosure. Studies have found that organizations with a more gender-diverse corporate board and executive team are more likely to have higher levels of ESG disclosure [25–27,29,32–36]. This finding is because gender diversity increases the likelihood of organizational transparency and accountability. Gender-diverse teams are also more likely to create a culture of ethical conduct, which is adequate for effective ESG disclosure.

The firms' RBVs support the positive relationship between gender diversity and corporate ESG disclosure. The RBV is a managerial framework used to assess the strategic resources a company can use to gain a sustainable competitive edge. This approach, championed by Barney [50], suggests firms differ due to their heterogeneous resources and, thus, different strategies. It directs attention to internal resources to determine those that can give a competitive advantage and maintain it. This theoretical framework suggests that organizations have access to both tangible and intangible resources that can be leveraged to create value. In the case of gender diversity, organizations can draw on the knowledge and perspectives of a wide range of individuals to create value. This diversity, in turn, can lead to improved corporate ESG disclosure.

In order to delve deeper into the significance of gender diversity in corporate ESG reporting within the RBV framework, this research concentrated on the proportion of women serving on a company's board of directors and the ratio of female executives to the total executive count. These factors determine the degree to which organizations harness and exploit the valuable asset of gender diversity. The firm's RBV perspective reinforces the positive correlation between gender diversity and corporate ESG reporting. The RBV is an organizational model employed to evaluate the strategic resources a company can utilize to achieve a lasting competitive advantage. This theoretical model posits that companies diverge due to their disparate resources and subsequent strategies, directing focus toward internal resources to identify those capable of providing a competitive edge and sustaining it. This theoretical approach implies that organizations possess both tangible and intangible resources that can be employed to generate value. In the context of gender diversity, organizations can benefit from the insights and viewpoints of a diverse group of individuals to create value. This paper's results, within the RBV framework's scope, suggest that gender diversity can be deemed a valuable asset for organizations, particularly regarding enhancing corporate ESG reporting. Companies with gender-balanced teams are more likely to cultivate an environment of ethical behavior, which, consequently, can result in improved corporate ESG reporting.

## 5. Conclusions

We tested the relationship between ESG disclosure scores and women's representation at the board and executive level while considering potential confounding factors. To address the possibility that other internal and external factors may influence ESG disclosure scores, we included control variables such as the company's annual return on equity, total debt ratio, and the natural logarithm of total assets as a proxy measurement of the firm's size. By incorporating these control variables, we tried to isolate the effects of women's representation on the board and executive level while accounting for the potential impact of other factors on the dependent variables. This approach helps obtain more meaningful results and provides a more accurate assessment of the relationship between ESG disclosure scores and gender diversity.

However, despite the rigorous statistical analysis, including control variables, we acknowledge that establishing causality in such complex relationships is challenging. It is important to note that gender diversity is only a narrow dimension of the broader diversity concept. Other diversities, such as nationality, race, age, professional background, and others, may also drive our results if they correlate with gender diversity. Firms that embrace a more comprehensive vision of diversity may also experience higher levels of ESG disclosure, suggesting that our findings partially reflect the impact of broader diversity measures on ESG disclosure.

The study's results provide evidence of a positive relationship between gender diversity and ESG disclosure scores, but it does not necessarily imply causation. Other unobserved factors, such as corporate culture, country-specific regulations, and industry norms, could also contribute to the observed relationship. Given the potential influence of such unobserved factors, including other diversity dimensions, on ESG disclosure, it represents an opportunity for further studies to differentiate the impact of different aspects of diversity on corporate ESG disclosure. By examining these relationships, researchers can better understand the underlying mechanisms and identify the most effective strategies for promoting ESG disclosure.

While our study presents strong evidence of a positive relationship between women's representation at the board and executive level and increased ESG disclosure scores, it is essential to remain cautious in attributing their upsurge solely to women's representation. Further research, including longitudinal studies and more comprehensive analyses of various contributing factors, including different dimensions of diversity, may strengthen the confidence in the claim that women's representation leads to improved ESG disclosure.

This article uses the RBV as its primary conceptual and analytical framework to explain the positive relationship between gender diversity and corporate ESG disclosure. By leveraging diverse teams' unique resources and capabilities, firms can enhance their ESG disclosure efforts and achieve better overall performance. This study underscores the importance of implementing policies that promote gender diversity at executive and board levels to ensure the improvement of corporate ESG disclosure. Governments should use this evidence to enforce policies promoting women's empowerment in the corporate world, ultimately leading to improved corporate ESG disclosure. Such policies may also rely on a growing literature review about the benefits of women's empowerment at the corporate level. Such benefits include enhanced corporate social responsibility [51,52], superior board strategic control & development [53], increased public disclosure [54,55], and better corporate social [56,57] performance. The benefits also comprise higher analysts' earnings forecast accuracy [58], and a lower likelihood of engaging in mergers and acquisitions [59]. Our results provide scientific evidence of an additional practical benefit of policies that foster corporate gender diversity represented by an enhanced ESG-related corporate disclosure performance. Such a benefit may result in an improved corporate value [60] without the potential adverse effects of imposing mandatory ESG disclosure regulations [61].

The limitations of this academic article include the analyzed data, which is only from 2016 to 2020 and may not accurately reflect the current situation. The study only focuses on publicly traded companies, so the results may not apply to privately held companies. Furthermore, the article does not provide any information on the impact of women's empowerment on firms' actual ESG-related performance, only on the amount of ESG-related disclosure. Therefore, more research is needed to understand the real impact of women's empowerment on firms' ESG-related performance. Lastly, the article does not consider any other external factors that could influence the results, such as the legal and regulatory environment in each country or the implementation of other initiatives to promote corporate gender diversity. Therefore, further research should consider these factors to draw more accurate conclusions.

This paper makes a unique contribution to the literature by being the first to analyze the effects of women's empowerment on ESG disclosure using a globally representative sample. However, the research can be further extended by focusing on the specifics of the research question. For instance, further research could be conducted to determine the effects of women's empowerment on ESG disclosure in different regions. It could also focus on the differences in impact between public and private companies. Research could also determine the effects of specific policies or initiatives to increase gender diversity on ESG disclosure. Additionally, the study could be conducted to explore the impact of different types of ESG disclosure metrics on the effects of women's empowerment. These are only a few of the many possible research extensions that could be conducted to explore further the impact of women's empowerment on corporate ESG disclosure.

Further research could explore how women's empowerment may affect corporate ESG disclosure. For example, studies could explore how women's empowerment affects corporate decision-making and how that decision-making impacts ESG disclosure. Additionally, analyses could be conducted to explore the differences in impact between initiatives aimed at increasing gender diversity and those aimed at increasing diversity in general. Additionally, studies could be undertaken to examine the effects of women's empowerment on different types of ESG disclosure metrics. Moreover, analyses could be conducted to determine the impact of varying board and executive representation levels on ESG disclosure. Finally, studies could also be extended to explore the effects of women's empowerment on the financial performance of publicly traded companies. Such studies could examine the degree to which women's empowerment has a positive or negative effect on a company's financial performance and the extent to which this relationship is mediated by ESG disclosure. This research could benefit investors and other stakeholders in the corporate world, as it would provide insight into the potential returns associated with investing in companies with gender-diverse boards and executive teams.

**Author Contributions:** All the authors participated equally in all phases of the research reported in the article, including conceptualization, methodology, software use, validation, formal analysis, investigation, resources, data curation, writing—original draft preparation, writing—review and editing, visualization, supervision, and project administration. All authors have read and agreed to the published version of the manuscript.

**Funding:** This research received no external funding.

**Institutional Review Board Statement:** Not applicable.

**Informed Consent Statement:** Not applicable.

**Data Availability Statement:** The study used only accessible secondary public data sources.

**Conflicts of Interest:** The authors declare no conflict of interest.

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
