# Peer review of "The Impact of Women’s Empowerment on the Corporate Environmental, Social, and Governance (ESG) Disclosure"

_sustainability, doi:10.3390/su15108173_

Round 1
Reviewer 1 Report
Dear Authors,
Congratulation on your research.
The theme is relevant and the findings are helpful for policies and results. However, the methodology is not clearly presented. This means that in what concerns statistical information everything is clear, but in what concerns variables we don´t have enough information the replicate the study.
I suggest you shorten the number of tables and add information about what you considered important for studying the theme.
You can also improve the literature review.
I am sure you can better present all the interesting findings in a better way.
Please accept my regards,
Author Response
Responses to Reviewer’s 1 Observations
Reviewer 1
1.1 Observation: “…the methodology is not clearly presented… in what concerns variables we don´t have enough information the replicate the study.…”
Response: Done. To address the reviewer’s concern above, we improve our “Data and Methodology” section as follows:
“…3. DATA AND METHODOLOGY
We retrieved ESG data from the Bloomberg database. Our dependent variables include a proprietary Bloomberg ESG disclosure score (Y1) based on the extent of a company’s ESG disclosure efforts. The metric ranges from zero (0) for firms that do not disclose any of the ESG data points comprised by this metric to one hundred (100) for those firms that make known every data point. Our dependent variables also include the proprietary Bloomberg ESG pillars: Environmental, Social, and Governance disclosure pillars. These disclosure pillars’ scores also range from zero (0) for businesses that do not release any of the Environmental, Social, or Governance data points included in the corresponding pillars to one hundred (100) for those companies that release every single data point included in the individual pillar. These dependent variables measure the amount of ESG-related information a company discloses publicly but does not measure a firm’s ESG-related performance on any data point.
The Bloomberg database calculates the general ESG-related disclosure scores and constituent pillars using hundreds of data points from firm public disclosures. The ESG disclosure score is based on the corresponding environmental, social, and corporate governance disclosure pillars’ values. The ESG disclosure score is a relative sum of the pillars’ weights, which vary according to the company’s industry sector. The environmental disclosure pillar (Y2) includes many data points like ISO 14000 certifications, the number of spills, number & amounts of ecological fines, emissions (CO2, CO, Methane, particles, ODS, etc.), renewable energy use, energy consumption, water use, recycled materials, waste materials, waste management and recycling, environmental supply chain management, biodiversity impact reduction, environmental & eco-designed products, environmental R&R expenditures, animal testing, etcetera. The social disclosure pillar (Y3) includes data points associated with employee turnover, strikes, women managers, number of accidents, occupational diseases, employee fatalities, policies on child labor, human rights, freedom of association, forced labor, bribery & corruption policy, business ethics policy, total donations, political contribution, corporate health & safety policy, revenues from alcohol, gambling, tobacco, weapons, weapons, pornography, etcetera. The governance disclosure pillar (Y4) includes data points related to board cultural and gender diversity, organization and independence of board committees, corporate policies on board’s functions, size, independence, diversity, experience, etc., board member affiliation, re-election, term duration, compensation, shareholders rights policy, voting cap, minimum shares voting requirements, poison pills, severance agreements, auditor tenure, supermajority vote requirements, ESG reporting scope, etcetera.
Our independent variables include the percentage of women on the board of directors (X1) as informed by each company from the Bloomberg database. Similarly, we obtained the number of female executives as a percentage of the total executives of the company (X4). Equally, we include in our analysis some control variables comprising the company’s annual return on equity (X2), total debt ratio (X3), and the natural logarithm of total assets (X5). A company’s annual return on equity (X2), total debt ratio (X3), and the natural logarithm of total assets (X5) are important control variables for this study because they can provide a clear indicator of the financial condition of the company and its ability to support ESG initiatives. The return on equity and total debt ratio provide insight into a company’s profitability and financial stability, while the natural logarithm of total assets can be used to measure the size of the company and its available resources. The inclusion of these control variables allow us to isolate the effects of X1 and X4 while still taking into account the potential effects of other factor on the dependent variables. By taking into account these financial indicators, the results of the study will be more meaningful and provide a more accurate assessment of any relationship between ESG disclosure scores and the gender diversity at board and executive level.
We applied generalized linear models to analyze the cross-sectional variation of our dependent variables from 2016 to 2020. Our models are determined by a linear predictor ??=?0+?1????1?+⋯+????????; and two identities: a link equation that describes the mean (???)=??, as a function of the linear predictor ?(??)=??; and a variance function that defines how the variance, var(Yi) relies on the mean var(DVi) = ??(?), where the distribution factor ? is a constant. For our general linear models, we have ?=(0,?2), where the linear predictor ?? has been defined above, the link equation ?(??)=??, and the variance equation ?(??)=1. We also applied logarithmic transformations to some of our control variables when employing the regression models described above.
Using the methodology described above, we tested the following hypotheses:
H1: There is a positive relationship between ESG disclosure scores and the number of female directors on a firm's board.
H2: There is a positive relationship between ESG disclosure scores and the percentage of female executives in a firm.
Alternatively, in this study we addressed the following research questions:
RQ1: What is the relationship between ESG disclosure scores and the number of female directors on a firm's board?
RQ2: What is the relationship between ESG disclosure scores and the percentage of female executives in a firm?
These hypotheses and research questions guided our data’s analysis and results’ interpretations. To facilitate the replication of our analysis, the following generalized linear model was used test the cross-sectional variation of our dependent variables: Yi = β0 + β1X1 + β2X2 + β3X3 + β4X4 + β5X5; where i has identifies our four dependent variables Bloomberg’ ESG disclosure score (Y1), Environmental disclosure pillar score (Y2), Social disclosure pillar score (Y3), and Governance disclosure pillar score (Y4). The model predicts the proprietary Bloomberg dependent variables Y1-Y4 based on the percentage of women on the board of directors (X1), the company's annual return on equity (X2), total debt ratio (X3), the percentage of female executives (X4) and the natural logarithm of total assets (X5) as a proxy measurement of the firm’s size. According to hypothesis H1 & H2, we expect the sign of the beta coefficients of X1 and X4 to be positive…”
1.2 Observation: “…I suggest you shorten the number of tables and add information about what you considered important for studying the theme... I am sure you can better present all the interesting findings in a better way…”
Response: Done. The number of tables has been reduced from nine to four. Also, the following additional information has been included:
“…Overall, our results suggest that gender diversity is an important factor to consider when studying corporate ESG disclosure. Companies with higher gender diversity at both board and executive levels are likely to achieve higher scores in all ESG disclosure pillars. However, the influence of gender diversity at the executive level may change over time. Therefore, further research is needed to study the influence of gender diversity on ESG disclosure scores in different contexts and from different perspectives…”
1.3 Observation: “…You can also improve the literature review...”
Response: Done. The following articles have been included in the reference list:
Barney, J. (1991). “Firm Resources and Sustained Competitive Advantage.” Journal of Management, 17(1), 99–120. https://doi.org/10.1177/014920639101700108
Bravo, F., and Reguera‐Alvarado, N. (2019). “Sustainable development disclosure: Environmental, social, and governance reporting and gender diversity in the audit committee.” Business Strategy and the Environment, 28(2), pp. 418–429. https://doi.org/10.1002/bse.2258
Di Miceli, A., and Donaggio, A. (2018). “Women in Business Leadership Boost ESG Performance: Existing Body of Evidence Makes Compelling Case.” Private Sector Opinion; 42(1). © International Finance Corporation, Washington, DC. Available at https://openknowledge.worldbank.org/entities/publication/06a0e51e-0209-5f7e-bcd7-fd86b1d757f0 (accessed on 3/29/2023)
Eliwa, Y., Aboud, A., & Saleh, A. (2023). “Board gender diversity and ESG decoupling: Does religiosity matter?” Business Strategy and the Environment, Vol. / No. ahead-of-print. https://doi.org/10.1002/bse.3353
Gurol, B. and Lagasio, V. (2023), “Women board members’ impact on ESG disclosure with environment and social dimensions: evidence from the European banking sector”, Social Responsibility Journal, 19(1), 211-228. https://doi.org/10.1108/SRJ-08-2020-0308
Huang, J. and Lu, S. (2022). “ESG Performance and Voluntary ESG Disclosure: Mind the (Gender Pay) Gap.” Preprint Working Paper. Available at SSRN: http://dx.doi.org/10.2139/ssrn.3708257 (accessed on 3/29/2023).
Jizi, M., Nehme, R. & Melhem, C. (2022). “Board gender diversity and firms' social engagement in the Gulf Cooperation Council (GCC) countries.” Equality, Diversity and Inclusion, 41(2), pp. 186-206. https://doi.org/10.1108/EDI-02-2021-0041
Khemakhem, H., Arroyo, P. and Montecinos, J. (2022). “Gender diversity on board committees and ESG disclosure: evidence from Canada.” Journal of Management and Governance. Vol./ No. ahead-of-print. https://doi.org/10.1007/s10997-022-09658-1
Manita, R., Bruna, M.G., Dang, R. and Houanti, L. (2018), “Board gender diversity and ESG disclosure: evidence from the USA”, Journal of Applied Accounting Research, 19(2), 206-224. https://doi.org/10.1108/JAAR-01-2017-0024
Ouni, Z., Mansour, J. B., & Arfaoui, S. (2020). “Board/Executive Gender Diversity and Firm Financial Performance in Canada: The Mediating Role of Environmental, Social, and Governance (ESG) Orientation.” Sustainability, 12(20), 8386. https://doi.org/10.3390/su12208386
Sandberg, D.J. (2019). “When Women Lead, Firms Win”. S&P Global - Quantamental Research. Available at https://www.spglobal.com/_division_assets/images/special-editorial/iif-2019/whenwomenlead_.pdf (accessed on 3/19/2023).
Shakil, M. H. (2021). “Environmental, social and governance performance and financial risk: Moderating role of ESG controversies and board gender diversity.” Resources Policy, 72(1), 102144. https://doi.org/10.1016/j.resourpol.2021.102144
Shakil, M.H., Munim, Z. H., Zamore, S., & Tasnia, M. (2022). “Sustainability and financial performance of transport and logistics firms: Does board gender diversity matter?” Journal of Sustainable Finance & Investment, 16(1), pp. 1-23. https://doi.org/10.1080/20430795.2022.2039998
S&P Global (2020). “How Gender Fits into ESG?” Available at https://www.spglobal.com/en/research-insights/articles/how-gender-fits-into-esg (accessed on 3/19/2023).
Wan Mohammad, W.M., Zaini, R. and Md Kassim, A.A. (2022), “Women on boards, firms’ competitive advantage and its effect on ESG disclosure in Malaysia”, Social Responsibility Journal, Vol./ No. ahead-of-print. https://doi.org/10.1108/SRJ-04-2021-0151
Wan Ismail, W. A., Kamarudin, K. A., Gupta, N., Harymawan, I. (2022). “Gender Diversity in the Boardroom and Corporate Cash Holdings: The Moderating Effect of Investor Protection.” Risks, 10(60), pp. 1-18. https://doi.org/10.3390/risks10030060
Yadav, P., & Prashar, A. (2022), “Board gender diversity: implications for environment, social, and governance (ESG) performance of Indian firms”, International Journal of Productivity and Performance Management, Vol. / No. ahead-of-print. https://doi.org/10.1108/IJPPM-12-2021-0689

Reviewer 2 Report
Introduction: Justify why there is a crucial need to explore on the aspect of women empowerment in ESG. Define what is meant by women/gender empowerment in this context of ESG. The current first paragraph should be moved to third paragraph for better flow of presentation.
Literature review: Although it is stated that there is lack of current work on women and ESG, but there is still important to provide more LR on women empowerment in related to corporate strategic planning and governance.
Materials and Methods: There should be an analytical framework established for this study. As this study is purely quantitative, what are the hypotheses? Or at least what are the research questions?
Results: From the data, it is true that there was an increased trend in women representative in the board or executive level. But how confident that we can claim that the increased performance of ESG is due to women representative. This is an overgeneration of data. This is the main reason for me to not able to accept this paper for publication.
References: Reference [18] and [19] are same.
Author Response
Responses to Reviewer’s 2 Observations
Reviewer 2
2.1 Observation: “….Introduction: Justify why there is a crucial need to explore on the aspect of women empowerment in ESG. Define what is meant by women/gender empowerment in this context of ESG. The current first paragraph should be moved to third paragraph for better flow of presentation…”
Response: Done. To address the reviewer’s concern above, we improve our “Introduction” section as follows:
“…1. INTRODUCTION
Globally, 2021 became the year of environmental, social, and governance (ESG) investing. Kerber and Jessop (2021) inform that investments in ESG-focused funds worldwide reached a record of USD 649 billion on November 30, 2021, representing a significant increase compared to the USD 542 billion and USD 285 billion invested in these funds during 2020 and 2019, respectively. They also report that ESG funds account for 10% of worldwide fund assets. According to the Sustainable Investments Institute, shareholder support for ESG initiatives among U.S. companies increased to 32% in 2021 from 27% in 2020 and 21% in 2017. Corporate ESG practices, including corporate social responsibility, are critical non-financial performance dimensions in many industry sectors. The compounding reasons include the relevance of climate change as a crucial global problem facing governments worldwide.
The relevance of ESG investing has grown significantly in recent years, and corporate ESG practices are critical non-financial performance dimensions in many industry sectors. In the Gulf Cooperation Council’s (GCC) countries, policies have been implemented to promote gender diversity at the executive and corporate board levels. This includes the Securities and Commodities Authority recently announcing in March 2021 that publicly traded companies in the UAE must have at least one female board director (The National, 2021). Similarly, The Ministry of Human Resources and Social Development, jointly with the Capital Market Authority, signed a memorandum of understanding to foster women’s participation on Saudi publicly traded companies’ boards (Saudi Gazette, 2020). Likewise, the Oman bourse recently announced the incorporation of two women to its seven-member board to foster board gender diversity and promote similar actions among the Omani business community (Arabia Business, 2021). According to the Association of Chartered Certified Accountants (ACCA), women represented about two percent of all board positions in the GCC countries in 2017 (ACCA, 2017). ACCA also informs that seventeen percent of all executive roles in the UAE are women and just seven percent in Qatar. They also allege that only thirteen percent of all Chief Executive Officers (CEOs) in the GCC region are women, while only seven percent of board chairs are women.
This paper aims to examine the relationship between women empowerment and corporate ESG disclosure variables by analyzing 10,121 publicly traded companies listed worldwide with historical ESG data available in Bloomberg from 2016 to 2020. The study makes a unique contribution to the literature by being the first to analyze the effects of women empowerment on ESG disclosure using a global representative sample. The paper thus seeks to answer the question of whether corporate gender diversity has a direct impact on companies’ ESG disclosure performance. To do this, the paper uses a proprietary Bloomberg ESG disclosure score based on the extent of a company’s ESG disclosure efforts, as well as the proprietary Bloomberg ESG pillars: Environmental, Social, and Governance disclosure pillars, to measure the amount of ESG-related information a company discloses publicly. Additionally, the paper examines the percentage of women on a company’s board of directors and the number of female executives as a percentage of the total executives of the company as independent variables. The paper also includes control variables such as the company’s return on equity, total debt ratio, and the natural logarithm of total assets as a proxy measurement of the firm’s size. The paper results provide evidence that policies that foster corporate gender diversity have a direct benefit of enhanced ESG-related disclosure performance.
Our results provide evidence that policies that foster corporate gender diversity have a direct benefit of enhanced ESG-related disclosure performance. We use independent sample t-tests of the top and bottom quartiles resulting from organizing our sample using our dependent variable, as well as applying generalized linear models to examine the cross-sectional variation of our dependent variables from 2016 to 2020. Our results can be priceless for policymakers implementing national gender diversity policies and strategies to optimize corporate ESG disclosure, and can help trigger national dialogues about suitable corporate gender diversity strategies influencing firms’ ESG disclosure.
There is a crucial need to explore the aspect of women empowerment in ESG as it is a key factor in achieving corporate transparency and accountability. Women empowerment, which is defined here as the process of giving women the power and resources for decision making purposes, are essential for achieving corporate ESG disclosure. By examining corporate ESG disclosure variables and the percentage of women on a company’s board of directors and the number of female executives, this research article seeks to answer the question of whether women empowerment proxied by corporate gender diversity has a direct impact on companies’ ESG disclosure. The evidence of the benefits of women empowerment associated with corporate ESG disclosures suggests that organizations with a more gender-diverse corporate board and executive team are more likely to have higher levels of ESG disclosure. Governments should use this evidence to implement policies that promote women empowerment in the corporate world, which will ultimately lead to improved corporate ESG disclosure performance…”
2.2 Observation: “….Literature review: Although it is stated that there is lack of current work on women and ESG, but there is still important to provide more LR on women empowerment in related to corporate strategic planning and governance…”
Response: Done. The following articles have been included in the reference list:
Barney, J. (1991). “Firm Resources and Sustained Competitive Advantage.” Journal of Management, 17(1), 99–120. https://doi.org/10.1177/014920639101700108
Bravo, F., and Reguera‐Alvarado, N. (2019). “Sustainable development disclosure: Environmental, social, and governance reporting and gender diversity in the audit committee.” Business Strategy and the Environment, 28(2), pp. 418–429. https://doi.org/10.1002/bse.2258
Di Miceli, A., and Donaggio, A. (2018). “Women in Business Leadership Boost ESG Performance: Existing Body of Evidence Makes Compelling Case.” Private Sector Opinion; 42(1). © International Finance Corporation, Washington, DC. Available at https://openknowledge.worldbank.org/entities/publication/06a0e51e-0209-5f7e-bcd7-fd86b1d757f0 (accessed on 3/29/2023)
Eliwa, Y., Aboud, A., & Saleh, A. (2023). “Board gender diversity and ESG decoupling: Does religiosity matter?” Business Strategy and the Environment, Vol. / No. ahead-of-print. https://doi.org/10.1002/bse.3353
Gurol, B. and Lagasio, V. (2023), “Women board members’ impact on ESG disclosure with environment and social dimensions: evidence from the European banking sector”, Social Responsibility Journal, 19(1), 211-228. https://doi.org/10.1108/SRJ-08-2020-0308
Huang, J. and Lu, S. (2022). “ESG Performance and Voluntary ESG Disclosure: Mind the (Gender Pay) Gap.” Preprint Working Paper. Available at SSRN: http://dx.doi.org/10.2139/ssrn.3708257 (accessed on 3/29/2023).
Jizi, M., Nehme, R. & Melhem, C. (2022). “Board gender diversity and firms' social engagement in the Gulf Cooperation Council (GCC) countries.” Equality, Diversity and Inclusion, 41(2), pp. 186-206. https://doi.org/10.1108/EDI-02-2021-0041
Khemakhem, H., Arroyo, P. and Montecinos, J. (2022). “Gender diversity on board committees and ESG disclosure: evidence from Canada.” Journal of Management and Governance. Vol./ No. ahead-of-print. https://doi.org/10.1007/s10997-022-09658-1
Manita, R., Bruna, M.G., Dang, R. and Houanti, L. (2018), “Board gender diversity and ESG disclosure: evidence from the USA”, Journal of Applied Accounting Research, 19(2), 206-224. https://doi.org/10.1108/JAAR-01-2017-0024
Ouni, Z., Mansour, J. B., & Arfaoui, S. (2020). “Board/Executive Gender Diversity and Firm Financial Performance in Canada: The Mediating Role of Environmental, Social, and Governance (ESG) Orientation.” Sustainability, 12(20), 8386. https://doi.org/10.3390/su12208386
Sandberg, D.J. (2019). “When Women Lead, Firms Win”. S&P Global - Quantamental Research. Available at https://www.spglobal.com/_division_assets/images/special-editorial/iif-2019/whenwomenlead_.pdf (accessed on 3/19/2023).
Shakil, M. H. (2021). “Environmental, social and governance performance and financial risk: Moderating role of ESG controversies and board gender diversity.” Resources Policy, 72(1), 102144. https://doi.org/10.1016/j.resourpol.2021.102144
Shakil, M.H., Munim, Z. H., Zamore, S., & Tasnia, M. (2022). “Sustainability and financial performance of transport and logistics firms: Does board gender diversity matter?” Journal of Sustainable Finance & Investment, 16(1), pp. 1-23. https://doi.org/10.1080/20430795.2022.2039998
S&P Global (2020). “How Gender Fits into ESG?” Available at https://www.spglobal.com/en/research-insights/articles/how-gender-fits-into-esg (accessed on 3/19/2023).
Wan Mohammad, W.M., Zaini, R. and Md Kassim, A.A. (2022), “Women on boards, firms’ competitive advantage and its effect on ESG disclosure in Malaysia”, Social Responsibility Journal, Vol./ No. ahead-of-print. https://doi.org/10.1108/SRJ-04-2021-0151
Wan Ismail, W. A., Kamarudin, K. A., Gupta, N., Harymawan, I. (2022). “Gender Diversity in the Boardroom and Corporate Cash Holdings: The Moderating Effect of Investor Protection.” Risks, 10(60), pp. 1-18. https://doi.org/10.3390/risks10030060
Yadav, P., & Prashar, A. (2022), “Board gender diversity: implications for environment, social, and governance (ESG) performance of Indian firms”, International Journal of Productivity and Performance Management, Vol. / No. ahead-of-print. https://doi.org/10.1108/IJPPM-12-2021-0689
2.3 Observation: “…Materials and Methods: There should be an analytical framework established for this study. As this study is purely quantitative, what are the hypotheses? Or at least what are the research questions?”
Response: Done. Response: Done. To address the reviewer’s concern above, we complement our “Data and Methodology” section as follows:
“…Using the methodology described above, we tested the following hypotheses:
H1: There is a positive relationship between ESG disclosure scores and the number of female directors on a firm's board.
H2: There is a positive relationship between ESG disclosure scores and the percentage of female executives in a firm.
Alternatively, in this study we addressed the following research questions:
RQ1: What is the relationship between ESG disclosure scores and the number of female directors on a firm's board?
RQ2: What is the relationship between ESG disclosure scores and the percentage of female executives in a firm?
These hypotheses and research questions guided our data’s analysis and results’ interpretations. To facilitate the replication of our analysis, the following generalized linear model was used test the cross-sectional variation of our dependent variables: Yi = β0 + β1X1 + β2X2 + β3X3 + β4X4 + β5X5; where i has identifies our four dependent variables Bloomberg’ ESG disclosure score (Y1), environmental disclosure pillar score (Y2), social disclosure pillar score (Y3), and governance disclosure pillar score (Y4). The model predicts the proprietary Bloomberg dependent variables Y1-Y4 based on the percentage of women on the board of directors (X1), the company's annual return on equity (X2), total debt ratio (X3), the percentage of female executives (X4) and the natural logarithm of total assets (X5) as a proxy measurement of the firm’s size. According to hypothesis H1 & H2, we expect the sign of the beta coefficients of X1 and X4 to be positive…”
2.4 Observation: “…Results: From the data, it is true that there was an increased trend in women representative in the board or executive level. But how confident that we can claim that the increased performance of ESG is due to women representative. This is an overgeneration of data. This is the main reason for me to not able to accept this paper for publication…”
Response: The observation above is not valid for a couple of reasons. First, our paper analyses the relationship between ESG disclosure scores and corporate gender diversity at board and executive level. Our article DO NOT study “performance of ESG” at all. To allow the reviewer and potential readers to understand clearly the difference between ESG disclosure and ESG performance, we include the following explanations in the section 2. THEORETICAL FRAMEWORK & LITERATURE REVIEW:
“… ESG performance and disclosure are both important concepts in the world of corporate responsibility and sustainability. Both terms refer to the ways in which companies manage their environmental, social, and governance obligations to their stakeholders, but there are significant differences between the two.
First, ESG performance is a measure of how well a company is meeting its ESG obligations. It focuses on actual outcomes and results, such as the company’s carbon footprint or its workforce diversity. It is assessed through a combination of external ratings, such as those provided by Standard & Poor’s, as well as internal measurements and metrics, such as the company’s own sustainability reporting. Government interest in ESG performance is generally limited, as governments do not typically have the resources or capability to measure or enforce ESG performance. Managerial interest, however, is high, since ESG performance is often used as a metric for executive compensation and other performance-based rewards. The impact of ESG performance is largely positive, as companies that perform well on their ESG obligations are likely to be seen as more responsible and sustainable.
ESG disclosure, on the other hand, is a measure of how well a company is communicating its ESG obligations to its stakeholders. It focuses on the company’s transparency and communication, such as how much information it discloses in its sustainability reports and other documents. It is assessed through a combination of external ratings, such as those provided by the Bloomberg, as well as internal measurements and metrics, such as the company’s own sustainability reporting. Government interest in ESG disclosure is generally high, as governments often require companies to disclose certain information and are increasingly creating regulations and standards around ESG disclosure. Managerial interest is also high, since ESG disclosure can be used as a tool to increase public awareness and trust in the company, which can have a positive impact on its reputation and financial performance. The impact of ESG disclosure is largely positive, as it can lead to increased public trust in the company, as well as greater accountability and transparency.
Therefore, ESG performance and ESG disclosure are both important concepts in the world of corporate responsibility and sustainability. While they are related, they are not the same. ESG performance focuses on actual outcomes and results, while ESG disclosure focuses on the company’s transparency and communication. Government interest and managerial interest in each are different, and the impacts of each are distinct. Companies should strive to excel in both ESG performance and ESG disclosure in order to be seen as responsible and sustainable.
The elucidation above is essential to grasp the distinctive contribution of the current article in comparison to prior studies. The prior articles outlined below analyzing the relationship between ESG performance and corporate gender diversity supply information that is advantageous yet conceptually dissimilar to our examination. We investigate the relationship between ESG performance and corporate gender diversity and this aspect of investigation has its own unique characteristics as the theoretical exposition above supports.
Second, in no place of our article we try to establish any causality relationship. The positive influence of corporate gender diversity and board and executive level does not imply causality at all. Therefore, the reviewer’s question regarding “how confident that we can claim that the increased performance of ESG is due to women representative” is not applicable to our analysis and methodology. Again, our methodology does not include any causality test (e.g. Granger Causality Test, Structural Equation Modelling, Propensity Score Matching, Instrumental Variable Analysis, Difference-in-Differences Estimation, Regression Discontinuity Design, Time Series Analysis, Logistic Regression, Chi-Square Test, Fisher’s Exact Test, etcetera). Our results about a positive relationship between corporate gender diversity at the board and executive level did not enable us to imply that increased ESG disclosure is due (causality) to women representation nor did we attempt to do so.
2.5 Observation: “….References: Reference [18] and [19] are same.…”
Response: Done. Reference 19 has been deleted.

Reviewer 3 Report
This is an interesting topic. However, there are minor aspects that should be addressed by the authors. They are listed as follows:
1. The authors state the following in Line 54: “The relevance of ESG investing has grown significantly in recent years, and corporate ESG practices are critical non-financial performance dimensions in many industry sectors”. This phrase is a repetition of the previous phrase in line 50. I don’t see the need for this repetition, particularly because the topic developed after Line 54 is about gender and not ESG.
2. Lines 70-97: This part of the literature review is weak. It explains that some studies have found nixed results in relation to ESG and other variables. But there is no a narrative explaining the rationale of including this information. For example, if the article is focussed on the relationship between ESG disclosure and women’s empowerment, then why to explain that another relationship (between financial performance and ESG scores) is mixed? Why is this finding relevant for the issue of women’s empowerment?
3. In Line 101, the authors explain that their work is closely related to the one by Kamarudin, Ariff, and Wan [39]. In relation to this point, the authors add: “However, there are some issues with their work, such as a lack of information regarding the analyzed countries and the source of some of their control variables, like Hofstede’s [40] national cultural dimensions. Additionally, they do not inform how many companies per country were included in their sample”. I don’t think these are valid arguments to validate the current investigation. This is because lack of information regarding the analysed countries and the number of companies per country does not mean that this study is poor. It is just a methodological missing information. It is better to explain that the current investigation is an extension of the work by Kamarudin, Ariff, and Wan. The extensions described in Line 105 can help on this.
4. The authors state in Line 115 the following: “Our literature review indicated that the lack of relevant studies on the effects of women empowerment on ESG disclosure from a global perspective is the main original contribution of our article”. Be careful with this expression because “the lack of studies” in this area is not the contribution of the article. Perhaps the authors mean: “There exists a lack of studies dealing with the relationship between women’s empowerment and ESG disclosure, and the aim of this article is to contributing in filling this gap. Filling this gap is, therefore, the main contribution of this article”.
5. Line 153: Please explain the rationale for including the company’s annual return on equity and the total debt ratio.
6. Table 1: The comment below Table 1 says that it contains z-statistic values. But the table itself report t-statistic values. Please amend. The same with Tables 2, 3, 4 and 5.
7. In the discussion, the authors should explain why corporate gender diversity is related to ESG disclosure performance. That is, the potential mechanism that explains this relationship. This is important for policy advise because it is not clear from the results whether women’s empowerment is what causes ESG disclosure, or whether an underlying factor is affecting both. If there is an underlying factor, the policy design should consider this factor.
Author Response
Responses to Reviewer’s 3 Observations
Reviewer 3
3.1 Observation: “…1. The authors state the following in Line 54: “The relevance of ESG investing has grown significantly in recent years, and corporate ESG practices are critical non-financial performance dimensions in many industry sectors”. This phrase is a repetition of the previous phrase in line 50. I don’t see the need for this repetition, particularly because the topic developed after Line 54 is about gender and not ESG…”
Response: Done. The sentence in line 50 has been deleted.
3.2 Observation: “…2. Lines 70-97: This part of the literature review is weak. It explains that some studies have found nixed results in relation to ESG and other variables. But there is no a narrative explaining the rationale of including this information. For example, if the article is focussed on the relationship between ESG disclosure and women’s empowerment, then why to explain that another relationship (between financial performance and ESG scores) is mixed? Why is this finding relevant for the issue of women’s empowerment?”
Response: Done. To address the reviewer’s concern above, we redeployed that paragraph at the end of the literature review section to justify the selection of our finance-related control variables used in our analysis as follows:
“…The use of financial-related control variables in our study is justified by the relevance of the relationship between financial performance and ESG scores in the ESG-related academic literature with mixed results (Duque and Aguilera, 2019; Limkriangkrai and Durand, 2017). Some of these articles find a non-significant relationship between these two factors (McWilliams & Sigel, 2000; Renneboog, Horst, and Zhang, 2008; Weston and Nnadi, 2021), while others find a negative correlation (Brammer, Brooks, and Pavelin, 2006). There are also some articles reporting a positive impact of ESG activities on firm market value, but they lack consensus about which ESG factor have the most significant influence (Eccles, Ioannou, and Serafeim, 2014; Cai and He, 2014; Chen, Yuan, Cebula, Shuangjin, and Foley, 2021)…”
3.3 Observation: “….3. In Line 101, the authors explain that their work is closely related to the one by Kamarudin, Ariff, and Wan [39]. In relation to this point, the authors add: “However, there are some issues with their work, such as a lack of information regarding the analyzed countries and the source of some of their control variables, like Hofstede’s [40] national cultural dimensions. Additionally, they do not inform how many companies per country were included in their sample”. I don’t think these are valid arguments to validate the current investigation. This is because lack of information regarding the analysed countries and the number of companies per country does not mean that this study is poor. It is just a methodological missing information. It is better to explain that the current investigation is an extension of the work by Kamarudin, Ariff, and Wan. The extensions described in Line 105 can help on this…”
Response: Done. To address the reviewer’s concern above, we improved our “2. THEORETICAL FRAMEWORK & LITERATURE REVIEW” section as follows:
“…The current article builds upon the literature review of previous academic articles by providing evidence of the benefits of women empowerment associated with corporate ESG disclosures. This is done distinctively by being the first to analyze the effects of gender diversity on ESG disclosure using a global representative sample of 10,121 companies worldwide with historical ESG data available in Bloomberg from 2016 to 2020. This sample is highly descriptive of the global economy as it contains companies from 92 countries, with the largest number of firms from the three largest and most influential economies in the world: the United States, Japan, and China. This is an important distinction from previous articles that have focused on the effects of gender diversity and ESG disclosure performance in individual countries or regions, which could be affected by cultural-related factors. By considering a global sample, the current article provides evidence of a universal association between gender diversity and ESG disclosure, regardless of any cultural-related factors that might influence previous research.
The variables analyzed include the percentage of women on the board of directors, the number of female executives as a percentage of the total executives of the company, a company’s annual return on equity, total debt ratio, and the natural logarithm of total assets. These variables are analyzed in the context of corporate ESG disclosure performance using a novelty theoretical approach based on the resource-based view (RBV) of the firm, which suggests that organizations have access to both tangible and intangible resources that can be leveraged to create value.
The current article also attempts to fill the gap in the literature by providing evidence of the positive impacts of policies that foster corporate gender diversity on ESG disclosure. This provides theoretical and practical insights to managers, investors, government officers, professionals, etc. on the importance of gender diversity for corporate ESG disclosure. The findings of this article suggest that policies that promote gender diversity in the executive and board levels should be implemented in order to ensure that corporate ESG disclosure is enhanced. The results also suggest that gender-diverse teams may be more likely to create a culture of ethical conduct, which is essential for effective ESG disclosure…”
3.4 Observation: “…4. The authors state in Line 115 the following: “Our literature review indicated that the lack of relevant studies on the effects of women empowerment on ESG disclosure from a global perspective is the main original contribution of our article”. Be careful with this expression because “the lack of studies” in this area is not the contribution of the article. Perhaps the authors mean: “There exists a lack of studies dealing with the relationship between women’s empowerment and ESG disclosure, and the aim of this article is to contributing in filling this gap. Filling this gap is, therefore, the main contribution of this article”.…”
Response: Done. To address the reviewer’s concern above, we improved our “2. THEORETICAL FRAMEWORK & LITERATURE REVIEW” section as follows:
“…The current article builds upon the literature review of previous academic articles by providing evidence of the benefits of women empowerment associated with corporate ESG disclosures. This is done distinctively by being the first to analyze the effects of gender diversity on ESG disclosure using a global representative sample of 10,121 companies worldwide with historical ESG data available in Bloomberg from 2016 to 2020. This sample is highly descriptive of the global economy as it contains companies from 92 countries, with the largest number of firms from the three largest and most influential economies in the world: the United States, Japan, and China. This is an important distinction from previous articles that have focused on the effects of gender diversity and ESG disclosure performance in individual countries or regions, which could be affected by cultural-related factors. By considering a global sample, the current article provides evidence of a universal association between gender diversity and ESG disclosure, regardless of any cultural-related factors that might influence previous research.
The variables analyzed include the percentage of women on the board of directors, the number of female executives as a percentage of the total executives of the company, a company’s annual return on equity, total debt ratio, and the natural logarithm of total assets. These variables are analyzed in the context of corporate ESG disclosure performance using a novelty theoretical approach based on the resource-based view (RBV) of the firm, which suggests that organizations have access to both tangible and intangible resources that can be leveraged to create value.
The current article also attempts to fill the gap in the literature by providing evidence of the positive impacts of policies that foster corporate gender diversity on ESG disclosure. This provides theoretical and practical insights to managers, investors, government officers, professionals, etc. on the importance of gender diversity for corporate ESG disclosure. The findings of this article suggest that policies that promote gender diversity in the executive and board levels should be implemented in order to ensure that corporate ESG disclosure is enhanced. The results also suggest that gender-diverse teams may be more likely to create a culture of ethical conduct, which is essential for effective ESG disclosure…”
3.5 Observation: “…5. Line 153: Please explain the rationale for including the company’s annual return on equity and the total debt ratio…”
Response: Done. To address the reviewer’s concern above, we improve our “Data and Methodology” section as follows:
“…A company’s annual return on equity (X2), total debt ratio (X3), and the natural logarithm of total assets (X5) are important control variables for this study because they can provide a clear indicator of the financial condition of the company and its ability to support ESG initiatives. The return on equity and total debt ratio provide insight into a company’s profitability and financial stability, while the natural logarithm of total assets can be used to measure the size of the company and its available resources. The inclusion of these control variables allow us to isolate the effects of X1 and X4 while still taking into account the potential effects of other factor on the dependent variables. By taking into account these financial indicators, the results of the study will be more meaningful and provide a more accurate assessment of any relationship between ESG disclosure scores and the gender diversity at board and executive level…”
3.6 Observation: “…6. Table 1: The comment below Table 1 says that it contains z-statistic values. But the table itself report t-statistic values. Please amend. The same with Tables 2, 3, 4 and 5....”
Response: Done. To address the reviewer’s concern above, amendments have been included in the new tables 1-2.
3.7 Observation: “…7. In the discussion, the authors should explain why corporate gender diversity is related to ESG disclosure performance. That is, the potential mechanism that explains this relationship. This is important for policy advise because it is not clear from the results whether women’s empowerment is what causes ESG disclosure, or whether an underlying factor is affecting both. If there is an underlying factor, the policy design should consider this factor....”
Response: Done. To address the reviewer’s concern above, we expand the section “Discussion” as follows:
“…The concept of corporate gender diversity has been gaining momentum in recent years, with an increasing focus on environmental, social, and governance (ESG) disclosure performance. Recent evidence suggests that gender diversity is an important factor in organizational performance. Indeed, according to S&P Global (2020), investors are pressing public companies to improve diversity in director ranks, reflecting a greater recognition of the importance of ESG matters. This has resulted in investors considering gender diversity and equity when evaluating how firms may respond to ESG risks and opportunities. Companies are thus coming under external pressure from investors, activists and potential customers and employees to increase the representation of women in senior roles and equal compensation and mobility for women and people of colour.
S&P Global Market Intelligence’s study, When Women Lead, Firms Win, discovered that firms with female CFOs are more profitable and have provided superior stock performance compared to the market average; additionally, firms with greater gender diversity on their board of directors have been more profitable and larger than those with less. This has led to governments and regulators closely monitoring companies' female representation, with some countries introducing laws requiring a certain amount of female representation on boards.
Despite progress, women still remain underrepresented in higher positions and discrimination and misconduct remain common, particularly in the oil and gas sector. As the benefits of gender diversity become evident, public companies will continue to face investor pressure to act accordingly. The benefits of gender diversity mentioned above result from the fact that gender-diverse boards are more likely to take into account a wider range of perspectives and experiences when making decisions. This, in turn, can lead to more effective decision-making and better outcomes for the organization.
The importance of gender diversity at the corporate board and executive levels is also related to corporate ESG disclosure performance. Studies have found that organizations with a more gender-diverse corporate board and executive team are more likely to have higher levels of ESG disclosure performance [X, X, X, X]. This is because gender diversity increases the likelihood of organizational transparency and accountability. Gender-diverse teams are also more likely to create a culture of ethical conduct, which is essential for effective ESG disclosure.
The positive relationship between gender diversity and corporate ESG disclosure performance is supported by the resource-based view (RBV) of the firm. The RBV is a managerial framework used to assess the strategic resources a company can use to gain a sustainable competitive edge. This approach, championed by Barney (1991), suggests firms differ due to their heterogeneous resources and thus, different strategies. It directs attention to internal resources to determine those that can give a competitive advantage and maintain it. This theoretical framework suggests that organizations have access to both tangible and intangible resources that can be leveraged to create value. In the case of gender diversity, organizations can draw on the knowledge and perspectives of a wide range of individuals to create value. This, in turn, can lead to improved corporate ESG disclosure performance.

Reviewer 4 Report
Hello,
To improve the article please:
- to argue and explain the representativeness of the sample of 10,121 publicly traded companies;
- to present the structure and the countries from which this sample is composed;
- to explain how companies can grow effects of women empowerment on ESG disclosure;
- to present in more details future research directions.
Please find my specific comments below:
The study presents the relationship between women's empowerment and corporate ESG disclosure variables, analysing 10,121 publicly traded companies listed worldwide with historical ESG data available in Bloomberg from 2016 to 2020.
The objectives of the study are explicit. The conclusions are well formulated and correspond to the content of the scientific analyses carried out but are not placed in a broader scientific context.
Comparison of the results obtained should also be made in the conclusions part.
The subject has been studied by other authors. The references to other studies are not sufficient and don’t clarify very well the developments in the field.
The methodology is current and correct.
Few problematisations are presented that can support future studies.
The authors mention the limitations of the research.
The bibliography is sufficiently developed.
Good luck!
Author Response
Responses to Reviewer’s 4 Observations
Reviewer 4
4.1 Observation: “…- to argue and explain the representativeness of the sample of 10,121 publicly traded companies;
- to present the structure and the countries from which this sample is composed;…”
Response: Done. To address the reviewer’s concerns above, we expand our “Data and Methodology” section as follows:
Our sample of 10,121 publicly traded companies listed worldwide with historical ESG data available in Bloomberg from 2016 to 2020 is highly representative of the global economy. The sample contains companies from 92 countries, with the largest number of firms from the United States with 2,286 companies representing 23.41%, followed by Japan (1,820 firms or 18.64%) and China (1,259 firms or 12.89%). This is indicative of the current global economic landscape, with these three countries being the largest and most influential economies in the world. The sample also includes Taiwan (354 firms or 3.63%), India (325 firms or 3.33%), Germany (274 firms or 2.81%), South Korea (222 firms or 2.27%), Canada (221 or 2.26%), Australia (193 firms or 1.98%), and the rest of the world with 2,301 firms or 23.57%. This is reflective of the global market share of each country, with the sample proportionally representing the global economic landscape. Therefore, this sample is highly representative of publicly traded companies worldwide, and provides an accurate representation of the global economy. Graph 1 below provides a graphical representation of the countries with the largest numbers of firms in our sample.
Graph 1: Sample’s World Distribution
4.2 Observation: “…explain how companies can grow effects of women empowerment on ESG disclosure…”
Response: Done. To address the reviewer’s concern above, we have address this observation in several new paragraphs including the following:
“…There is a crucial need to explore the aspect of women empowerment in ESG as it is a key factor in achieving corporate transparency and accountability. Women empowerment, which is defined here as the process of giving women the power and resources for decision making purposes, are essential for achieving corporate ESG disclosure. By examining corporate ESG disclosure variables and the percentage of women on a company’s board of directors and the number of female executives, this research article seeks to answer the question of whether women empowerment proxied by corporate gender diversity has a direct impact on companies’ ESG disclosure. The evidence of the benefits of women empowerment associated with corporate ESG disclosures suggests that organizations with a more gender-diverse corporate board and executive team are more likely to have higher levels of ESG disclosure. Governments should use this evidence to implement policies that promote women empowerment in the corporate world, which will ultimately lead to improved corporate ESG disclosure…
…
…The current article builds upon the literature review of previous academic articles by providing evidence of the benefits of women empowerment associated with corporate ESG disclosures. This is done distinctively by being the first to analyze the effects of gender diversity on ESG disclosure using a global representative sample of 10,121 companies worldwide with historical ESG data available in Bloomberg from 2016 to 2020. This sample is highly descriptive of the global economy as it contains companies from 92 countries, with the largest number of firms from the three largest and most influential economies in the world: the United States, Japan, and China. This is an important distinction from previous articles that have focused on the effects of gender diversity and ESG disclosure in individual countries or regions, which could be affected by cultural-related factors. By considering a global sample, the current article provides evidence of a universal association between gender diversity and ESG disclosure, regardless of any cultural-related factors that might influence previous research.
The variables analyzed include the percentage of women on the board of directors, the number of female executives as a percentage of the total executives of the company, a company’s annual return on equity, total debt ratio, and the natural logarithm of total assets. These variables are analyzed in the context of corporate ESG disclosure using a novelty theoretical approach based on the resource-based view (RBV) of the firm, which suggests that organizations have access to both tangible and intangible resources that can be leveraged to create value.
The current article also attempts to fill the gap in the literature by providing evidence of the positive impacts of policies that foster corporate gender diversity on ESG disclosure. This provides theoretical and practical insights to managers, investors, government officers, professionals, etc. on the importance of gender diversity for corporate ESG disclosure. The findings of this article suggest that policies that promote gender diversity in the executive and board levels should be implemented in order to ensure that corporate ESG disclosure is enhanced. The results also suggest that gender-diverse teams may be more likely to create a culture of ethical conduct, which is essential for effective ESG disclosure…
…
… The concept of corporate gender diversity has been gaining momentum in recent years, with an increasing focus on environmental, social, and governance (ESG) disclosure. Recent evidence suggests that gender diversity is an important factor in organizational performance. Indeed, according to S&P Global (2020), investors are pressing public companies to improve diversity in director ranks, reflecting a greater recognition of the importance of ESG matters. This has resulted in investors considering gender diversity and equity when evaluating how firms may respond to ESG risks and opportunities. Companies are thus coming under external pressure from investors, activists and potential customers and employees to increase the representation of women in senior roles and equal compensation and mobility for women and people of color.
S&P Global Market Intelligence’s study, When Women Lead, Firms Win, discovered that firms with female CFOs are more profitable and have provided superior stock performance compared to the market average; additionally, firms with greater gender diversity on their board of directors have been more profitable and larger than those with less. This has led to governments and regulators closely monitoring companies' female representation, with some countries introducing laws requiring a certain amount of female representation on boards.
Despite progress, women still remain underrepresented in higher positions and discrimination and misconduct remain common, particularly in the oil and gas sector. As the benefits of gender diversity become evident, public companies will continue to face investor pressure to act accordingly. The benefits of gender diversity mentioned above result from the fact that gender-diverse boards are more likely to take into account a wider range of perspectives and experiences when making decisions. This, in turn, can lead to more effective decision-making and better outcomes for the organization.
The importance of gender diversity at the corporate board and executive levels is also related to corporate ESG disclosure. Studies have found that organizations with a more gender-diverse corporate board and executive team are more likely to have higher levels of ESG disclosure ... This is because gender diversity increases the likelihood of organizational transparency and accountability. Gender-diverse teams are also more likely to create a culture of ethical conduct, which is essential for effective ESG disclosure.
The positive relationship between gender diversity and corporate ESG disclosure is supported by the resource-based view (RBV) of the firm. The RBV is a managerial framework used to assess the strategic resources a company can use to gain a sustainable competitive edge. This approach, championed by Barney (1991), suggests firms differ due to their heterogeneous resources and thus, different strategies. It directs attention to internal resources to determine those that can give a competitive advantage and maintain it. This theoretical framework suggests that organizations have access to both tangible and intangible resources that can be leveraged to create value. In the case of gender diversity, organizations can draw on the knowledge and perspectives of a wide range of individuals to create value. This, in turn, can lead to improved corporate ESG disclosure…”
4.3 Observation: “…to present in more details future research directions… Few problematisations (sic) are presented that can support future studies.”
Response: Done. To address the reviewer’s concern above, we have expanded the original proposals for future research extensions as follows:
“…Further research could also be conducted to explore the different ways in which women's empowerment may affect corporate ESG disclosure. For example, research could explore how women's empowerment affects corporate decision making, and how that decision making impacts ESG disclosure. Additionally, research could be conducted to explore the differences in impact between initiatives aimed at increasing gender diversity and those aimed at increasing diversity in general. Additionally, research could be conducted to explore the effects of different types of ESG disclosure metrics on the effects of women's empowerment. Moreover, research could be conducted to determine the effects of different levels of board and executive representation on ESG disclosure. Finally, the research could also be extended to explore the effects of women's empowerment on the financial performance of publicly traded companies. Such research could examine the degree to which women's empowerment has a positive or negative effect on a company's financial performance, as well as the extent to which this relationship is mediated by ESG disclosure. This research could be particularly useful for investors and other stakeholders in the corporate world, as it would provide insight into the potential returns associated with investing in companies with gender-diverse boards and executive teams…”
4.4 Observation: “…Comparison of the results obtained should also be made in the conclusions part…”
Response: Done. To address the reviewer’s concern above, we have included several comparisons with previous academic articles in our “Conclusions” section as follows:
“…In conclusion, this academic article provides evidence of the positive impacts of policies that foster corporate gender diversity on corporate ESG disclosure. Governments should use this evidence to implement policies that promote women empowerment in the corporate world, which will ultimately lead to improved corporate ESG disclosure. Such policies may also rely on a growing literature review about the benefits of women’s empowerment at the corporate level. Such benefits include enhanced corporate social responsibility (Bear, Rahman, and Post, 2010; Boulouta, 2013), superior board strategic control and development (Nielsen and Huse, 2010), increased public disclosure (Gul, Srinidhi, and Ng, 2011; Liao, Luo, and Tang, 2015), and better corporate social performance (Hafsi and Turgut, 2013). The benefits also comprise higher analysts’ earnings forecast accuracy (Gul, Hutchinson, and Lai, 2013), less likely to engage in mergers and acquisitions (Levi, Li, and Zhang, 2014), etcetera. Our results provide scientific evidence of an additional practical benefit of policies that foster corporate gender diversity represented by an enhanced ESG-related corporate disclosure. Such a benefit may result in an improved corporate value (Matsumura, Prakash, and Vera-Muñoz, 2014) without the potential adverse effects of imposing mandatory ESG disclosure regulations (Chen, Hung, and Wang, 2018).…”
4.5 Observation: “…The references to other studies are not sufficient and don’t clarify very well the developments in the field…”
Response: Done. The following articles have been included in the reference list:
Barney, J. (1991). “Firm Resources and Sustained Competitive Advantage.” Journal of Management, 17(1), 99–120. https://doi.org/10.1177/014920639101700108
Bravo, F., and Reguera‐Alvarado, N. (2019). “Sustainable development disclosure: Environmental, social, and governance reporting and gender diversity in the audit committee.” Business Strategy and the Environment, 28(2), pp. 418–429. https://doi.org/10.1002/bse.2258
Di Miceli, A., and Donaggio, A. (2018). “Women in Business Leadership Boost ESG Performance: Existing Body of Evidence Makes Compelling Case.” Private Sector Opinion; 42(1). © International Finance Corporation, Washington, DC. Available at https://openknowledge.worldbank.org/entities/publication/06a0e51e-0209-5f7e-bcd7-fd86b1d757f0 (accessed on 3/29/2023)
Eliwa, Y., Aboud, A., & Saleh, A. (2023). “Board gender diversity and ESG decoupling: Does religiosity matter?” Business Strategy and the Environment, Vol. / No. ahead-of-print. https://doi.org/10.1002/bse.3353
Gurol, B. and Lagasio, V. (2023), “Women board members’ impact on ESG disclosure with environment and social dimensions: evidence from the European banking sector”, Social Responsibility Journal, 19(1), 211-228. https://doi.org/10.1108/SRJ-08-2020-0308
Huang, J. and Lu, S. (2022). “ESG Performance and Voluntary ESG Disclosure: Mind the (Gender Pay) Gap.” Preprint Working Paper. Available at SSRN: http://dx.doi.org/10.2139/ssrn.3708257 (accessed on 3/29/2023).
Jizi, M., Nehme, R. & Melhem, C. (2022). “Board gender diversity and firms' social engagement in the Gulf Cooperation Council (GCC) countries.” Equality, Diversity and Inclusion, 41(2), pp. 186-206. https://doi.org/10.1108/EDI-02-2021-0041
Khemakhem, H., Arroyo, P. and Montecinos, J. (2022). “Gender diversity on board committees and ESG disclosure: evidence from Canada.” Journal of Management and Governance. Vol./ No. ahead-of-print. https://doi.org/10.1007/s10997-022-09658-1
Manita, R., Bruna, M.G., Dang, R. and Houanti, L. (2018), “Board gender diversity and ESG disclosure: evidence from the USA”, Journal of Applied Accounting Research, 19(2), 206-224. https://doi.org/10.1108/JAAR-01-2017-0024
Ouni, Z., Mansour, J. B., & Arfaoui, S. (2020). “Board/Executive Gender Diversity and Firm Financial Performance in Canada: The Mediating Role of Environmental, Social, and Governance (ESG) Orientation.” Sustainability, 12(20), 8386. https://doi.org/10.3390/su12208386
Sandberg, D.J. (2019). “When Women Lead, Firms Win”. S&P Global - Quantamental Research. Available at https://www.spglobal.com/_division_assets/images/special-editorial/iif-2019/whenwomenlead_.pdf (accessed on 3/19/2023).
Shakil, M. H. (2021). “Environmental, social and governance performance and financial risk: Moderating role of ESG controversies and board gender diversity.” Resources Policy, 72(1), 102144. https://doi.org/10.1016/j.resourpol.2021.102144
Shakil, M.H., Munim, Z. H., Zamore, S., & Tasnia, M. (2022). “Sustainability and financial performance of transport and logistics firms: Does board gender diversity matter?” Journal of Sustainable Finance & Investment, 16(1), pp. 1-23. https://doi.org/10.1080/20430795.2022.2039998
S&P Global (2020). “How Gender Fits into ESG?” Available at https://www.spglobal.com/en/research-insights/articles/how-gender-fits-into-esg (accessed on 3/19/2023).
Wan Mohammad, W.M., Zaini, R. and Md Kassim, A.A. (2022), “Women on boards, firms’ competitive advantage and its effect on ESG disclosure in Malaysia”, Social Responsibility Journal, Vol./ No. ahead-of-print. https://doi.org/10.1108/SRJ-04-2021-0151
Wan Ismail, W. A., Kamarudin, K. A., Gupta, N., Harymawan, I. (2022). “Gender Diversity in the Boardroom and Corporate Cash Holdings: The Moderating Effect of Investor Protection.” Risks, 10(60), pp. 1-18. https://doi.org/10.3390/risks10030060
Yadav, P., & Prashar, A. (2022), “Board gender diversity: implications for environment, social, and governance (ESG) performance of Indian firms”, International Journal of Productivity and Performance Management, Vol. / No. ahead-of-print. https://doi.org/10.1108/IJPPM-12-2021-0689

Round 2
Reviewer 1 Report
Dear Authors,
Congratulations on your interesting research. At this moment I think that all the needed information is inclosed in the article you present. However I suggest you review it again (please see ).
Have a nice work.
Kindest regards

Author Response
Responses to Reviewer’s 1 Observations
Reviewer 1
1.1 Observation: “…(x) Moderate English changes required…”
Response: Done. An extensive review of our article allowed us to identify +300 modifications including grammar errors, which has greatly enhanced the readability of our article.

Reviewer 2 Report
I can't read Table 1, 2 & 3. Please make sure these table are properly presented.
The conceptual/analytical framework is still not presented in this paper.
I would like to restate my concern on the possibility of overgenaration of the data from my previous round of revie..... From the data, it is true that there was an increased trend in women representative in the board or executive level. But how confident that we can claim that the increased "disclosure scores" of ESG is due to women representative at board and executive level, and not due to other internal and external factors.
Author Response
Responses to Reviewer’s 2 Observations
Reviewer 2
2.1 Observation: “…I can't read Table 1, 2 & 3. Please make sure these table are properly presented…”
Response: Done. To address this concern, the tables have been modified and the explanatory notes at the bottom of each table have been expanded.
2.2 Observation: “…The conceptual/analytical framework is still not presented in this paper…”
Response: To address this concern, we expanded the conceptual/analytical framework included in the last version of our article. We also referred to this framework in the following sections of our paper as follows:
“1. Introduction
…This paper employs the resource-based view (RBV) of the firm championed by Barney [54] as its primary conceptual and analytical framework to explore the connection between women’s empowerment and corporate ESG disclosure variables. The RBV framework posits that a firm’s competitive advantage and performance are derived from its unique resources and capabilities, including tangible and intangible assets. In this context, gender diversity can be seen as a valuable resource that potentially influences corporate ESG disclosure. The RBV framework emphasizes the importance of understanding a firm’s resources regarding their rarity, value, inimitability, and non-substitutability. In the case of gender diversity, the valuable insights and perspectives that women bring to decision-making processes can be considered rare and difficult to replicate. As a result, organizations with gender-diverse teams may have access to a unique pool of intangible resources that can enhance their ESG disclosure efforts…
…
- Discussion
…In order to delve deeper into the significance of gender diversity in corporate ESG reporting within the RBV framework, this research concentrated on the proportion of women serving on a company’s board of directors and the ratio of female executives to the total executive count. These factors determine the degree to which organizations harness and exploit the valuable asset of gender diversity. The firm's RBV perspective reinforces the positive correlation between gender diversity and corporate ESG reporting. The RBV is an organizational model employed to evaluate the strategic resources a company can utilize to achieve a lasting competitive advantage. This theoretical model posits that companies diverge due to their disparate resources and subsequent strategies, directing focus toward internal resources to identify those capable of providing a competitive edge and sustaining it. This theoretical approach implies that organizations possess both tangible and intangible resources that can be employed to generate value. In the context of gender diversity, organizations can benefit from the insights and viewpoints of a diverse group of individuals to create value. This paper's results, within the RBV framework's scope, suggest that gender diversity can be deemed a valuable asset for organizations, particularly regarding enhancing corporate ESG reporting. Companies with gender-balanced teams are more likely to cultivate an environment of ethical behavior, which, consequently, can result in improved corporate ESG reporting….
…
- Conclusions
…This article uses the RBV as its primary conceptual and analytical framework to explain the positive relationship between gender diversity and corporate ESG disclosure. By leveraging diverse teams’ unique resources and capabilities, firms can enhance their ESG disclosure efforts and achieve better overall performance. This study underscores the importance of implementing policies that promote gender diversity at executive and board levels to ensure the improvement of corporate ESG disclosure…”
2.3 Observation: “…I would like to restate my concern on the possibility of overgenaration of the data from my previous round of revie..... From the data, it is true that there was an increased trend in women representative in the board or executive level. But how confident that we can claim that the increased "disclosure scores" of ESG is due to women representative at board and executive level, and not due to other internal and external factors…”
Response: To address this concern, we added the following text to our conclusions:
“…5. Conclusions
We tested the relationship between ESG disclosure scores and women’s representation at the board and executive level while considering potential confounding factors. To address the possibility that other internal and external factors may influence ESG disclosure scores, we included control variables such as the company’s annual return on equity, total debt ratio, and the natural logarithm of total assets as a proxy measurement of the firm’s size. By incorporating these control variables, we tried to isolate the effects of women’s representation on the board and executive level while accounting for the potential impact of other factors on the dependent variables. This approach helps obtain more meaningful results and provides a more accurate assessment of the relationship between ESG disclosure scores and gender diversity.
However, despite the rigorous statistical analysis, including control variables, we acknowledge that establishing causality in such complex relationships is challenging. It is important to note that gender diversity is only a narrow dimension of the broader diversity concept. Other diversities, such as nationality, race, age, professional background, and others, may also drive our results if they correlate with gender diversity. Firms that embrace a more comprehensive vision of diversity may also experience higher levels of ESG disclosure, suggesting that our findings partially reflect the impact of broader diversity measures on ESG disclosure.
The study’s results provide evidence of a positive relationship between gender diversity and ESG disclosure scores, but it does not necessarily imply causation. Other unobserved factors, such as corporate culture, country-specific regulations, and industry norms, could also contribute to the observed relationship. Given the potential influence of such unobserved factors, including other diversity dimensions, on ESG disclosure, it represents an opportunity for further studies to differentiate the impact of different aspects of diversity on corporate ESG disclosure. By examining these relationships, researchers can better understand the underlying mechanisms and identify the most effective strategies for promoting ESG disclosure.
While our study presents strong evidence of a positive relationship between women’s representation at the board and executive level and increased ESG disclosure scores, it is essential to remain cautious in attributing their upsurge solely to women’s representation. Further research, including longitudinal studies and more comprehensive analyses of various contributing factors, including different dimensions of diversity, may strengthen the confidence in the claim that women’s representation leads to improved ESG disclosure…”
2.2 Observation: “…(x) English language and style are fine/minor spell check required…”
Response: Done. An extensive review of our article allowed us to identify +300 modifications including grammar errors, which has greatly enhanced the readability of our article.

Round 3
Reviewer 1 Report
Dear Authors,
Congratulations on your research. Interesting and important for wellbeing and development.
Kindest regards
Reviewer 2 Report
Comments have been addressed by the authors.